# Revisiting ocean carbon sequestration by direct injection: A global carbon budget perspective

Fabian Reith[1], David P. Keller[1], Andreas Oschlies[1]

[1]Geomar Helmholtz-Centre for Ocean Research Kiel, Düsternbrooker Weg 20, 24105 Kiel, Germany

*Correspondence to*: Fabian Reith (FReith@geomar.de)

**Abstract.**

In this study we look beyond the previously studied effects of oceanic $CO_2$ injections on atmospheric and oceanic reservoirs, and also account for carbon cycle and climate feedbacks between the atmosphere and the terrestrial biosphere. Considering

these additional feedbacks is important since backfluxes from the terrestrial biosphere to the atmosphere in response to reducing atmospheric $CO_2$ can further offset the targeted reduction. To quantify these dynamics we use an Earth-system model of intermediate complexity to simulate direct injection of $CO_2$ into the deep ocean as a means of emissions mitigation during a high $CO_2$ emission scenario. In three sets of experiments with different injection depths, we simulate a 100-year injection period of a total of 70 GtC and follow global carbon cycle dynamics over another 900 years. In additional

parameter perturbation runs, we varied the default terrestrial photosynthesis $CO_2$ fertilization parameterization by ± 50% in order to test the sensitivity of this uncertain carbon cycle feedback to the targeted atmospheric carbon reduction through direct $CO_2$ injections. Simulated seawater chemistry changes and marine carbon storage effectiveness are similar to previous studies. As expected, by the end of the injection period avoided emissions fall short of the targeted 70 GtC by 16% to 30% as a result of carbon cycle feedbacks and backfluxes in both land and ocean reservoirs. The target emissions reduction in the

parameter perturbation simulations is about 0.2% and 2% more at the end of the injection period and about 9% less to 1 % more at the end of the simulations, when compared to the unperturbed injection runs.

An unexpected feature is the effect of the model's internal variability of deep-water formation in the Southern Ocean, which, in some model runs, causes additional oceanic carbon uptake after injection termination relative to a control run without injection and therefore with slightly different atmospheric $CO_2$ and climate. These results of a model that has very low

internal climate variability illustrate that attribution of carbon fluxes and accounting for injected $CO_2$ may be very

challenging in the real climate system with its much larger internal variability.

## 1. Introduction

Anthropogenic $CO_2$ emissions have perturbed the natural carbon cycle [Archer et al., 2009]. With an average of 8.6

$\pm$ 0.4 GtC yr$^{-1}$ emitted from fossil-fuel burning and 0.8 $\pm$ 0.5 GtC yr$^{-1}$ from land-use change in the last decade (2003 – 2013)

[Le Quéré et al., 2014], global $CO_2$ emissions have continuously increased by about 2.5 % yr$^{-1}$ [Friedlingstein et al., 2014].

This trend continues to follow slightly above the trajectory of the highest emission scenario of the latest IPCC report (see

section 2.2), which makes it very difficult to keep global warming within the political 2°C guardrail [Peters et al., 2013], not

to speak of recent agreements to seriously consider an even more ambitious 1.5°C goal [UNFCCC, 2015]. The limited

success in reducing or even slowing down the increase in anthropogenic emissions through global climate accords [Rogelj et

al., 2010] has led to renewed interest in engineering measures that are intended to reduce atmospheric $CO_2$ concentrations

[e.g., Shepherd, 2009].

Marchetti [1977] proposed directly injecting $CO_2$ into the deep ocean, thus accelerating the oceanic uptake of

atmospheric $CO_2$, which happens naturally via invasion and subsequent dissolution of $CO_2$ into the surface waters, albeit at a

relatively slow rate limited by the sluggish ocean overturning circulation. On time scales of thousands of years, however, this

will result in most anthropogenic $CO_2$ ending up in the deep ocean. The idea behind direct $CO_2$ injection is to speed up this

slow natural process by directly depositing $CO_2$ in deep waters, some of which remain isolated from the atmosphere for

hundreds to thousands of years [DeVries and Primeau, 2011; their Figure 12], thereby preventing the $CO_2$ from having an

effect on the climate in the near future. This is fundamentally different from just avoiding emissions, because the $CO_2$ has

still been added to the carbon cycle and may leak out of the ocean and affect the climate and other carbon cycle pathways.

Over millennial time scales carbon from direct injection can simply be viewed as "delayed" emissions, in terms of

its climatic effect and fate, since the carbon cycle will eventually reach a chemical equilibrium (mainly an equilibrium

between the oceanic and atmospheric carbon reservoirs, although carbonate compensation and weathering feedbacks start

acting on time scales longer than 5,000 years [e.g., Zeebe, 2012]). However, on decadal to centennial time scales, carbon that

is sequestered via direct injection cannot simply be treated as "delayed emissions" because the injected carbon must take

fundamentally different pathways than those of carbon that is emitted directly into the atmosphere. Since these pathways operate on many different time scales and are partially controlled by climate feedbacks, it takes a considerable amount of time until the carbon cycle and climate reach the same state as if the emissions had just been delayed. This is because injecting $CO_2$ changes ocean chemistry internally and thus, will at some point affect ocean carbon uptake or outgassing, and hence the atmospheric $CO_2$ concentration: when water with chemical properties having been altered by the injection reaches the surface, the air-sea exchange of $CO_2$ is fundamentally altered compared to a situation where the carbon was just emitted into the atmosphere at a later date. By sequestering carbon in the ocean instead of emitting it into the atmosphere, one would also inadvertently change terrestrial carbon cycling compared to the situation where the carbon was emitted with some delay.

While direct injection of $CO_2$ is presently in conflict with the London Protocol and the Convention for the Protection of the Marine Environment of the North East Atlantic (OSPAR Convention) [Leung et al., 2014], and also because of the long timescales and global scales involved, models are ideally suited for investigating this method [Orr, 2004]. Modelling studies are also safer than actual experiments because the rapid changes in seawater chemistry that could occur if direct $CO_2$ injections were tested might potentially harm marine ecosystems. These risks may be especially high for deep-sea benthic environments such as cold-water corals and sponge communities, which are adapted to special living conditions and thus may have a low capacity to acclimatize to rapid pH changes in their environment [e.g. IPCC, 2005, WBGU, 2006; Gehlen et al., 2014]. In previous studies, relatively simple box models [e.g., Hoffert et al., 1979] and first-generation global ocean circulation models [Orr, 2004] were employed, focusing on the residence time of the injected $CO_2$ (i.e. effectiveness), as well as on changes in ocean chemistry [e.g., Orr et al., 2001; Orr 2004; Jain and Cao, 2005; IPCC, 2005; Ridgwell et al., 2011].

However, a more comprehensive assessment of the carbon sequestration and climate mitigation potential of direct injection also requires accounting for the changes in all ambient carbon fluxes resulting from carbon cycle and climate feedbacks [Mueller et al., 2004; Vichi et al., 2013].

In this study, which follows Orr et al. [2001] in the configuration of the $CO_2$ injection scenarios, we use an Earth system model of intermediate complexity and fully interactive carbon cycle to simulate the direct injection of $CO_2$ into the deep ocean at different depths under a high $CO_2$ emission scenario. Our main objective is to assess the long-term response of

the atmospheric, oceanic and terrestrial carbon pools to the targeted atmospheric reduction through a continuous 100-year injection of $CO_2$ at seven offshore sites with individual injection rates (0.1 GtC $yr^{-1}$ each) that are small compared to today's global $CO_2$ emissions. Although previous studies [e.g., Orr et al., 2001; Orr 2004] have looked at the effects of $CO_2$ injections on atmospheric and oceanic reservoirs, the carbon-cycle and climate feedbacks between the atmosphere and the terrestrial biosphere were not considered in those studies because their models used did not have a land component. Considering these feedbacks is important since simulations of other oceanic carbon sequestration methods have shown that backfluxes from the terrestrial biosphere to the atmosphere can partially offset any oceanic C uptake [Oschlies et al., 2010]. However, since the future strength of terrestrial carbon cycle feedbacks, such as the $CO_2$ fertilization effect, is of uncertain magnitude as atmospheric $CO_2$ changes [e.g., Matthews, 2007; IPCC, 2013, Hajima et al., 2014], we also conduct parameter perturbation simulations, in which the default $CO_2$ fertilization parameterization of the terrestrial photosynthesis model is varied by ±50%. This allows us to better understand how differences in the response of the terrestrial biosphere affect the targeted atmospheric carbon reduction during direct $CO_2$ injections. For our injection simulations we use a well-calibrated model that conserves carbon globally, features the pelagic carbonate chemistry and is run under a business as usual emission scenario. The model and emission forcing used are identical to the ones in the Climate Engineering modelling study by Keller et al. [2014].

## 2. Methodology

### 2.1 Model Description

The model used is version 2.9 of the University of Victoria Earth System Climate Model (UVic ESCM). It consists of four dynamically coupled components: a three-dimensional general circulation ocean model (Pacanowski, 1996), a dynamic-thermodynamic sea-ice model (Bitz and Lipscomb, 1999), a terrestrial model [Meissner et al., 2003], and a one-layer atmospheric energy-moisture balance model [based on Fanning and Weaver, 1996]. All components have a common horizontal resolution of 3.6° longitude x 1.8° latitude. The oceanic component has 19 vertical levels with thicknesses ranging from 50 m near the surface to 500 m in the deep ocean. Formulations of the air-sea gas exchange and seawater carbonate chemistry are based on the OCMIP abiotic protocol [Orr et al., 1999]. The terrestrial model of vegetation and carbon cycles

is based on the Hadley Center model TRIFFID [e.g., Matthews, 2007]. A more detailed description of the UVic model version used here is given in Keller et al. [2012] and Eby et al. [2013].

## 2.2 Experimental Design

The model has been spun-up for 10,000 years under preindustrial atmospheric and astronomical boundary conditions and run from 1765 to 2005 using historical fossil fuel and land-use carbon emissions (Keller et al., 2014). From the year 2006 to 2100 the model is forced with $CO_2$ emissions following the Representative Concentration Pathway (RCP) 8.5, which is a business-as-usual high $CO_2$ emission scenario. Subsequently, simulations follow the Extended Concentration Pathway (ECP) 8.5 emission scenario until the year 2500 [Meinshausen et al., 2011]. Thereafter, we keep emissions constant at 1.48 GtC $yr^{-1}$ until the end of the simulations in year 3020. Note that non-$CO_2$ greenhouse gases and anthropogenic aerosol forcing agents as well as emissions from land-use change are not considered in our simulations.

Continental ice sheets, volcanic forcing and astronomical boundary conditions are held constant to facilitate the experimental setting and analyses (e.g., to prevent confounding feedback effects) [Keller et al., 2014]. Parameterized geostrophic wind anomalies, which are a first-order approximation of dynamical feedbacks associated with changing winds in a changing climate (Weaver et al., 2001), are also applied.

Simulated $CO_2$ injections into different ocean regions are based on the Ocean Carbon Cycle Model Intercomparison Project (OCMIP) carbon sequestration protocols [see Orr et al., 2001; Orr 2004] to facilitate comparison of our model results to those of Orr et al. [2001] and Orr [2004]. For simplicity, we simulate the injection of $CO_2$ in an idealized manner by adding $CO_2$ directly to the dissolved inorganic carbon (DIC) pool [Orr, 2001], thus neglecting any gravitational effects and assuming that the injected $CO_2$ instantaneously dissolves into seawater and is transported quickly away from the injection point and distributed homogeneously over the entire model grid box with lateral dimensions of a few hundred kilometers and many tens of meters in the vertical direction. Consequently, the formation of $CO_2$ plumes or lakes as well as the potential risk of fast rising $CO_2$ bubbles are neglected [IPCC, 2005; Bigalke et al., 2008]. Furthermore, we do not investigate the effect of $CaCO_3$ sediments feedbacks in our experiments, although the dissolution of $CaCO_3$ sediments near or downstream of an injection site is expected to reduce outgassing and increase the residence time of the injected $CO_2$ [Archer et al., 1998].

To track the physical transport of the injected $CO_2$ and its transport pathways from the individual injection sites, injected carbon is added to seven site-specific diagnostic marker tracers. At the sea surface, we assume that these tracers have an instantaneous gas exchange with the atmosphere, i.e., as soon as the injected carbon reaches an ocean surface grid box, the value of the marker tracer in this surface ocean grid box is set to zero. The residence time of the injected $CO_2$ computed from this tracer approach (i.e. fraction retained, see below) thus, provides a conservative estimate of carbon stored to carbon injected, as it is unlikely that all of the injected carbon would instantly leave the ocean upon reaching a depth of 50 m. Furthermore, the fraction retained is not affected by changes in the Revelle Factor related to the invasion of anthropogenic $CO_2$ into the ocean.

In all of our injection simulations we subtract the amount of injected $CO_2$ from the emissions forcing, thus keeping the total global carbon inventory the same as in the respective control simulation without $CO_2$ injection. For the purpose of assessing how all ambient carbon fluxes affect the storage lifetime of the injected $CO_2$, it is essential to have the same carbon inventory in all of our simulations. Following Orr et al. [2001] and Orr [2004], seven injection sites are located in individual grid boxes near the Bay of Biscay (42.3°N, 16.2°W), New York (36.9°N, 66.6°W), Rio de Janeiro (27.9°S, 37.8°W), San Francisco (31.5°N, 131.4°W), Tokyo (33.3°N, 142.2°E), Jakarta (11.7°S, 102.6°E) and Mumbai (13.5°N, 63°E) (Fig. 1). Starting in the year 2020, the experimental simulations consist of two periods: 1) an initial 100 year period of simultaneous 0.1 GtC $yr^{-1}$ injections and 2) a continuation of the model simulations until year 3020 after stopping the injections at the end of year 2119. Separate injection (I) experiments following this protocol are conducted at three different depths, 850 m (*I-800*), 1600 m (*I-1500*), and 2900 m (*I-3000*). Hereafter, these are referred to as *With Emissions* simulations.

Following previous studies [e.g., Jain and Cao, 2005; Ridgwell et al., 2011] additional simulations are conducted to investigate how climate-change induced feedbacks affect the fate of injected $CO_2$. These simulations follow the same protocols described above, but with anthropogenic emissions forcing set to zero from the year 2020 until the end of the simulations (year 3020). Hereafter, these extreme scenarios are referred to as *Complete Mitigation* simulations. Note that since these simulations are forced with historical emissions and the RCP 8.5 scenario until year 2020, the model is not in steady state in 2020 and some climatic change occurs. Also, because the injected $CO_2$ is withdrawn from the atmosphere so that total carbon is conserved, the *Complete Mitigation* injection runs essentially have negative emissions of 0.7 GtC $yr^{-1}$.

To determine how long the injected carbon stays in the ocean, we follow the IPCC [2005] and calculate a fraction retained ($FR = M_o * M_i^{-1} * 100$), which is the percentage ratio between the total mass of the injected carbon that remains in the ocean ($M_o$, determined using the diagnostic marker tracer) and the total cumulative mass injected into the ocean ($M_i$) since the start of the injection period (year 2020). This metric accounts for the injected carbon atoms and does not include possible adjustments of fluxes of other carbon in the Earth system.

To assess the global carbon cycle response to the injections, we use another metric, the net fraction stored ($netFS = \Delta C_{ocean} * M_i^{-1} * 100$, in %) that measures total carbon reservoir changes. The *netFS* is defined as the ratio between the absolute change in globally integrated total oceanic carbon ($\Delta C_{ocean}$), relative to the *RCP 8.5 control run*, and the total cumulative mass injected into the ocean ($M_i$) since the start of the injection period. In contrast to *FR* that counts only the injected carbon atoms, *netFS* accounts for all potential feedbacks of carbon fluxes into and out of the ocean in response to the injection of $CO_2$ into the ocean.

To investigate if the targeted atmospheric carbon reductions in the *With Emissions* simulations differ from what would happen if $CO_2$ was never emitted (avoided emissions) or first emitted and subsequently removed from the atmosphere, e.g., via technology such as direct air capture (see section 3.4.1) [Lackner, 2009] with subsequent safe and permanent storage, presumably in geological reservoirs, we performed another simulation where the atmospheric $CO_2$ concentration was 0.7 GtC yr$^{-1}$ less than in the RCP 8.5 control run between the years 2020 and 2120. Hereafter, this simulation is referred to as the *Direct Air Capture* run.

As mentioned in the introduction, this modelling study of direct $CO_2$ injection into the deep ocean is the first one to include a land component in order to assess, in addition to the atmospheric and oceanic carbon reservoirs, the long-term response of the terrestrial carbon pool to the targeted atmospheric carbon reduction through direct $CO_2$ injections. Since there is a significant amount of uncertainty in how the terrestrial system responds to changing atmospheric $CO_2$ concentrations [Friedlingstein et al., 2006], we have chosen to conduct several simulations with different terrestrial parameter values, i.e., a perturbed parameter study, to better understand how the terrestrial system could potentially respond to and affect the carbon cycle during deep ocean $CO_2$ injections. The parameterization that we investigate is the $CO_2$ fertilization effect. The process of $CO_2$ fertilization is thought to stimulate terrestrial carbon uptake [e.g., Matthews, 2007].

This negative carbon cycle feedback results in reduced atmospheric $CO_2$ concentrations, and has likely accounted for a substantial portion of the historical terrestrial carbon sink [Friedlingstein et al., 2006]. Accordingly, it has direct relevance for the future trajectory of atmospheric $CO_2$ [IPCC, 2013] and thus for our targeted atmospheric carbon reduction of 70 GtC by the year 2120. However, the future strength of $CO_2$ fertilization in response to changing $CO_2$ is highly uncertain [e.g., Friedlingstein et al., 2006; Arora et al., 2013; Jones et al., 2013; Schimel et al., 2015]. In order to better quantify the role of $CO_2$ fertilization in the targeted atmospheric carbon reduction in the *With Emissions* simulations (section 3.4.3), we vary the $CO_2$ fertilization parameterization following the approach of Matthews [2007]. Thereby, we scale the $CO_2$ sensitivity of the terrestrial photosynthesis model by ± 50% ($CO_2$ fertilization = high / low) for repeated simulations that are otherwise identical to the RCP 8.5 control, I-800 and I-3000 runs. These variations scale the default strength of an increase in atmospheric $CO_2$ increase relative to pre-industrial levels that is used to calculate all processes in the canopy and leaf routines within the terrestrial photosynthesis model, leading to a respective increase or decrease in terrestrial gross primary productivity. This is achieved by adding the multiplicative parameter '$CO_2\_fert\_scale$' in the routine of the photosynthesis model and setting it to 1.5 for an increase of the $CO_2$ fertilization effect and to 0.5 for a respective decrease.

Hereafter, the perturbed control runs are referred to as *RCP 8.5 control*$_{CO2\_fert\_high}$ and *RCP 8.5 control*$_{CO2\_fert\_low}$. The perturbed injections runs are denoted as *I-800* $_{CO2\_fert\_high}$, *I-800*$_{CO2\_fert\_low}$, *I-3000* $_{CO2\_fert\_high}$ and *I-3000*$_{CO2\_fert\_low}$. We did not perform an *I-1500* run because an ocean deep convection event that occurred after the injection period (see section 3.4.2) would make it too difficult to evaluate the results. No additional spin-up is needed; since the $CO_2$ fertilization effect only happens when atmospheric $CO_2$ concentration begins to increase, e.g., from the pre-industrial period onward.

An overview of all conducted simulations with their anthropogenic forcing is shown in Table 1.

## 3. Results and Discussion

### 3.1 RCP 8.5 control simulation

The physical climate and biogeochemical cycles of the Earth System during the RCP 8.5 control simulation are in the same state as described in Keller et al. [2014]. Here, we briefly describe global carbon cycling during the control simulation so that comparisons can be made to the *With Emissions* simulations (section 3.4). Subsequently, we briefly outline the global carbon cycling of the perturbed control runs *RCP 8.5 control*$_{CO2\_fert\_high}$ and *RCP 8.5 control*$_{CO2\_fert\_low}$ for

comparing these simulations to the unperturbed control run and the respective injection experiments (section 3.4.3).

By the end of the simulation in year 3020, about 6,000 GtC have been added to the global carbon cycle. Consequently, atmospheric $CO_2$ has increased substantially in the *RCP 8.5 control run*, leading to a total atmospheric carbon content of about 4620 GtC at the end of the simulation (Figs. S1, 2 a).

By the end of the extended *RCP 8.5 control run* about 58 % of the emitted $CO_2$ remains in the atmosphere. The rest of the carbon has been taken up by oceanic and terrestrial reservoirs (Figs. 2 e, i). Oceanic carbon uptake is highest during the first few decades of the simulation, when emissions are highest, and then decreases thereafter (Fig. 2 c). The decrease in net oceanic carbon uptake is particularly caused by a reduction in the ocean buffering capacity [Prentice et al., 2001], leading to a decrease in ocean carbon uptake even under increasing atmospheric $CO_2$ levels; a response also seen in other model simulations [Zickfeldt et al., 2013].

Simulated terrestrial carbon uptake is initially high as well, but then declines rapidly, with the terrestrial reservoir becoming a source for atmospheric carbon in the year 2139 before leveling off at very little net exchange between the terrestrial reservoir and the atmosphere after about year 2280 (Fig. 2 g). The initial increase in total land carbon uptake is due to the simulated $CO_2$ fertilization effect on vegetation [Matthews, 2007]. However, as temperatures become higher, terrestrial net primary productivity (NPP) is reduced due to water stress. Moreover soil respiration increases with temperature until it eventually becomes the dominant processes, leading to a net loss of carbon from the terrestrial reservoir to the atmosphere. Projections of future net terrestrial carbon uptake or loss processes are highly uncertain (Carvalhais et al., 2014; Hagerty et al., 2014; van der Sleen et al., 2014; Sun et al., 2014), which is also reflected in the large variability between the CMIP5 (Coupled Model Intercomparison Project Phase 5) model results, with changes in terrestrial carbon budgets ranging from -0.97 to +2.27 GtC yr$^{-1}$ between 2006 and 2100 [Ahlström et al., 2012].

As expected, simulated terrestrial carbon uptake is higher in the *RCP 8.5 control$_{CO2\_fert\_high}$* simulation because NPP is higher (not shown), when compared to the standard *RCP 8.5 control run*, resulting in a percentage increase in terrestrial carbon of about 5% in the year 2120 and of about 3% at the end of the simulation (Figs. 2 i, j). However, terrestrial carbon uptake declines more rapidly than in the control run, which is due to a faster saturation of the $CO_2$ fertilization effect as well as higher soil respiration. Consequently, the terrestrial biosphere switches about 20 years earlier to a stronger net carbon

source (year 2121) before leveling off at very little net exchange between the terrestrial reservoir and the atmosphere after about year 2280 as occurring in the standard *control run* (Fig. 2 i).

Accordingly, the atmospheric carbon concentration in the *RCP 8.5 control$_{CO2\_fert\_high}$* is lower, when compared to the *RCP 8.5 control run*, although the trends are similar (Figs. 2 a, b). Compared to the extended *RCP 8.5 control run*, the extended *RCP 8.5 control$_{CO2\_fert\_high}$* ends with about 1% less atmospheric carbon (Figs. 2 a, b). The lower atmospheric carbon content in the *RCP 8.5 control$_{CO2\_fert\_high}$*, caused by the higher $CO_2$ fertilization effect, leads initially to a reduced carbon flux from the atmosphere to ocean (Fig. 2 c). By the year 2075, the carbon flux from the atmosphere to ocean is

slightly higher, when compared to the control run, as the carbon flux from atmosphere to land starts to decrease with increasing $CO_2$ emissions (Fig. 2 d, g). Thus, total oceanic carbon in the *control$_{CO2\_fert\_high}$* run stays below that of the control run with a percentage decrease of about 0.07% at the year 2120 and about 0.05% at the end of the simulation (Figs. 2 e, f).

        Global carbon cycling in the *RCP 8.5 control$_{CO2\_fert\_low}$* shows a similar response, although of opposite sign and higher magnitude (Fig. 2), which is for instance reflected in a percentage decrease in total land carbon of about 10% in the

235    year 2120 and about 7% at the end of the simulation, when compared to the control run (Figs. 2 i, j). This is caused by the decreased $CO_2$ fertilization effect, which results in less NPP and thus in lower soil respiration.

### 3.2 Changes in seawater chemistry

        Here, we compare the *With Emissions* simulations to the *RCP 8.5 control run* to assess injection-related seawater chemistry changes. By the final year of the injection period (year 2119), a total of 10 GtC is injected at each site (Fig. 1). The

respective increases in DIC and reductions in pH depend on how quickly the injected carbon is transported away from the injection sites by local ocean currents and mixing [see Orr, 2004]. Our model-predicted changes in DIC and pH at the injection sites (relative to the control run) are within the range of Orr [2004] (Table S1-2).

        Simulated ocean surface $pCO_2$ is lower in the $CO_2$ injection runs because of lower atmospheric $CO_2$ levels and the related decrease in air-sea carbon fluxes, which results in lower surface DIC concentrations and a slightly higher surface pH

(by 0.008 to 0.01 units compared to the control run).

### 3.3 Fractions retained

Here, we assess to which extent the simulated $CO_2$ injections are effective in keeping the injected carbon out of the atmosphere. This is described by the fractions retained (*FR*). The global *FR* of our *Complete Mitigation* and *With Emissions* simulations (Table 2) are within the full range of the GOSAC-OCMIP results [Orr et al., 2001; Orr, 2004]. The simulated *FR* (Table 2) increases with the depth of injection because it generally takes longer for deeper waters to again come into contact with the atmosphere, as also shown in previous studies [e.g., Caldeira et al. 2001; Orr et al., 2001; Orr, 2004; Jain and Cao, 2005].

By comparing the *With Emissions* and *Complete Mitigation* simulations at all depths, we can determine how climate change affects *FR*. As in previous studies, our results show that *FR* is enhanced by climate change [Jain and Cao, 2005; Ridgwell et al., 2011]. In the *With Emissions* simulations, values of *FR* are always higher than in the *Complete Mitigation* runs (Table 2). For *I-800* and *I-1500*, the *FR* increase due to climate change is largest in the Pacific, whereas for *I-3000*, Atlantic sites show the highest *FR* increase due to a larger ocean response to climate change (Table 2). However, in all simulations more of the injected carbon is retained in the Pacific compared to injections in other ocean basins.

We also assess whether the enhanced *FR* in our *With Emissions* simulations are affected by changes in the Atlantic Meridional Overturning Circulation (AMOC). Relative to preindustrial, which has a maximum AMOC intensity of 15.98 Sv, we find AMOC decreases by 8%, 29%, 40%, 34% in the years 2020, 2120, 2520, 3020, respectively in the *With Emissions* simulations. AMOC in the *Complete Mitigation* simulations, relative to preindustrial, shows smaller decreases of about 7.6%, 21%, 8.6%, 8.6% in the years 2020, 2120, 2520, 3020, respectively. These differences partially explain why *FR* is enhanced in the *With Emissions* simulations, since a reduced AMOC slows the transport of deep water masses and prolongs the time until they again come into contact with the atmosphere. As in other climate change studies [e.g., Doney, 2010; Bopp et al., 2013], we also find an increase in ocean stratification (not shown) in all respective basins in our *With Emissions* runs, relative to the *Complete Mitigation* runs, which has also led to reduced vertical mixing [Prentice et al., 2001] and increased *FR*. In contrast to Jain and Cao [2005], who found a higher *FR* mainly in the Atlantic, we find a higher *FR* in all basins (Table 2). This difference is likely related to the higher degree of climate change in our simulations since we use a higher $CO_2$ emissions scenario.

Model-predicted *FR* (Table 2) refers to the injected $CO_2$ alone (as accounted for by the diagnostic marker tracer) and does not account for how global carbon cycle feedbacks affect net ocean carbon storage. By comparing *FR* and net fraction stored (*netFS*, see section 2.2) for the *With Emissions* simulations, we find that net ocean C sequestration is less efficient than would be predicted from *FR* alone (Fig. 4 a) because of carbon cycle and climate feedbacks (Fig. 1). For *I-3000*, *netFS* is about 16% lower than *FR* at the end of the injection period (Table 2, Fig. 4 a).

These results show the importance of accounting for carbon cycle feedbacks when assessing the effectiveness of marine $CO_2$ injections. Interestingly, an exception occurs for the *I-1500* simulation from the last year of the injection period with a Southern Ocean deep convection event during which the ocean temporarily takes up more carbon than would be expected from the injections alone (Figs. 4 a, c, d). This event and its implications for carbon accounting are discussed in more detail in section 3.4.2.

## 3.4 Response of the Global Carbon Cycle

Here we first briefly show how the atmospheric carbon reduction, relative to the *RCP8.5 control run* (see section 3.1), differs between *With Emissions* simulations and the *Direct Air Capture* run. Subsequently, we investigate how carbon cycle and climate feedbacks affect the distribution of carbon between different reservoirs upon injection of $CO_2$ in the *With Emissions* simulations. To do so, we look at the absolute changes in carbon between the *With Emissions* simulations and *RCP 8.5 control run* during and after the injection period. Finally, we show how the perturbed injection runs, in which we scaled the default $CO_2$ fertilization parameterization of the terrestrial photosynthesis model [section 2.2], affect the targeted atmospheric carbon reduction as well as the other carbon reservoirs and fluxes in *I-800* and *I-3000* of the *With Emissions* simulations.

### 3.4.1 Response during injection period

In the *With Emissions* simulations and the *Direct Air Capture* run, the *globally injected carbon* denotes the targeted atmospheric carbon reduction. The *globally injected carbon* - in the absence of leakage and backfluxes - equals the oceanic carbon addition or atmospheric $CO_2$ removal of 70 GtC by the last year of the injection period (year 2119). As presented in

Figures 3, 4 b, the atmospheric carbon reduction during the injection period of the *With Emissions* simulations diverges quickly from the *globally injected carbon* trajectory.

This is explained by injected carbon leaking from the ocean back to the atmosphere and the response of atmosphere-to-land and atmosphere-to-ocean fluxes to the reduction in atmospheric carbon. The rapid divergence even for the deepest injection points where *FR* is high, points to carbon cycle and climate feedbacks, which are directly related to changes in atmospheric $CO_2$ concentrations (i.e. ocean-atmosphere $pCO_2$ differences and $CO_2$ fertilization effects) and changes in temperature. Other studies have also shown that these feedbacks occur and affect the size of the global carbon reservoirs (Arora et al., 2013). The curve progression of the atmospheric reduction in the *Direct Air Capture* run is very similar for *I-1500* and *I-3000*, which is due to the occurrence of most of the same carbon cycle and climate feedback mechanisms. However, due to no carbon injections in the *Direct Air Capture* run, the atmospheric reduction is higher as soon as injected carbon starts leaking in the *With Emissions* simulations as presented in Figure 3. In the UVic model (version 2.9), the atmospheric carbon reduction of the *Direct Air Capture* run (Fig. 3) can also be referred to as the true atmospheric carbon reduction target. Depending on depth of injection, this implies further that direct injection of $CO_2$ would not be able be 100% efficient and provide 100% of the true atmospheric reduction target on decadal to centennial timescales (Fig. 3). Due to the occurrence of an ocean deep convection event in the *Direct Air Capture* run after the year 2120 (see section 3.4.2), we cannot easily compare the *Direct Air Capture* run to the *With Emissions* simulations after the injection period.

While ocean feedbacks in response to $CO_2$ injection and reduced atmospheric $CO_2$ levels have been discussed extensively in previous studies [e.g. Orr 2004; IPCC, 2005, Ridgewell et al., 2011], we here additionally consider land feedbacks with the purpose of accounting for the entire Earth system's response to potential marine $CO_2$ injections.

By the last year of the injection period (year 2119), *I-800* shows the highest divergence from *globally injected carbon* (Fig. 4 c) with an atmospheric carbon reduction of only 48 GtC, which is 22 GtC less than targeted. Since from the marker tracer it is known that 25% (i.e. 17.8 GtC) of the injected $CO_2$ has leaked to the atmosphere (Table 2), C-cycle and temperature feedbacks must be responsible for the other 4.2 GtC that remained in the atmosphere. This remaining amount can partially be explained by the reduced $pCO_2$ difference between the atmosphere and the ocean, which leads to a smaller carbon flux into the ocean (Fig. 4 d). Plus, relative to the control run, there is a lower atmosphere-to-land carbon flux until

approximately the year 2075 (Fig. 4 f), leading to 1.2 GtC less total land carbon by the end of the injections (Fig. 4 e). After the injections start (year 2020), both NPP and soil respiration are lower in *I-800* than in the control run, leading to a maximum reduction in land carbon of about 4.2 GtC in year 2075 (Fig. 4 e). Thereafter, total land carbon in *I-800* increases. By the end of the injections in year 2120, the terrestrial carbon pools have taken up 1.2 GtC less than the control run without $CO_2$ injection.

Roughly similar patterns are found for injection simulations *I-1500* and *I-3000* during the injection period, although with less outgassing occurring for the deeper injections (Fig. 4 c), which led to a slightly larger reduction in terrestrial carbon uptake by the last year of the injection. Thus, the largest reduction in total atmospheric carbon with 60 GtC was found for *I-3000*, followed by *I-1500* with 58 GtC by the end of the injection period (Fig. 4 b).

Our results suggest that the terrestrial response due to the atmospheric carbon reduction is mainly governed by the reduced $CO_2$ fertilization effect on NPP and the temperature related decrease in soil respiration. Carbon cycle-climate feedbacks on land occur because the reduced atmospheric $CO_2$ concentration in the *With Emissions* simulations (Fig. 4 c) leads to a cooling in the global mean soil temperature by about 0.08°C to 0.1°C in the year 2119 relative to the control simulation, with the lowest reduction for *I-800* and the highest one for *I-3000*. Both fertilization and temperature feedbacks on the terrestrial biosphere act simultaneously, although our results indicate that the reduced $CO_2$ fertilization effect, which, in current models is the largest terrestrial carbon cycle feedback (Schimel et al., 2015), is the dominant one until the maximum reduction in land carbon around year 2075. Thereafter, the decrease in soil respiration leads to an increase in land carbon and becomes the dominant feedback.

Feedbacks from the terrestrial system to atmospheric $CO_2$ are among the largest uncertainties to projections of future climate change (Schimel et al., 2015). According to our analysis, these would impact our ability to predict the net carbon storage associated direct injection of $CO_2$ into the deep ocean.

The neglected effect of the $CaCO_3$ dissolution feedback in our injection experiments [see section 2.2] introduces another uncertainty with respect to the response of the global carbon cycle to direct $CO_2$ injections. Model simulations by Archer et al. [1998] have shown that $CaCO_3$ dissolution is sensitive to direct $CO_2$ injections throughout the Atlantic, but that it leads to only a slight impact on atmospheric $pCO_2$. However, a slightly modified trajectory of atmospheric $CO_2$ may, for

instance, further impact the terrestrial carbon pool and fluxes, and could result in different terrestrial responses as in our *With*

345 *Emissions* simulations. However, the comparison on how the marine $CaCO_3$ sediments feedback would affect global carbon cycling to the injections experiments without $CaCO_3$ sediments is the subject of future work and beyond the scope of this particular study.

### 3.4.2 Response after injection period

After the injections are stopped (end of year 2119), *I-800* shows a continuous outgassing of about 40 GtC until the

350 end of the simulation, which is represented by the steady divergence from the *globally injected carbon* (denoted as GIC in Figs. 4 b, c). As in the control simulation, the terrestrial system in *I-800* becomes a source of carbon between the years 2139 and 2280, although the flux is slightly lower because of lower atmospheric $CO_2$ and lower temperatures. Thus, the net effect is an increase in land carbon relative to the control simulation with a maximum of 3 GtC in the year 2239 (Fig. 4 e). Thereafter, total land carbon in I-800 converges towards that of the RCP 8.5 control run, but remains higher until the end of

355 the simulation (Fig. 4 e).

Unlike *I-800*, *I-3000* actually gets closer to the *globally injected carbon* trajectory after the end of the injection period until the year 2199, with about 64 GtC less total atmospheric carbon than in the control simulation, compared to about 60 GtC at the end of the injection period in year 2119 (Fig. 4 b). This is a result of the reduced carbon flux from the atmosphere to the ocean, relative to the RCP 8.5 control run (Fig. 4 d), with only about 4 GtC leaving the ocean by year

2199. Moreover, the land turns from a sink into a net source of $CO_2$ in year 2139 (Fig. 4 f). Subsequently, *I-3000* shows a steady outgassing of the injected $CO_2$ from the year 2199 until the end of the simulation (Fig. 4 e), with little change in the terrestrial carbon pool (Fig. 4 f). The processes that govern changes in terrestrial carbon in *I-3000* are the same as for *I-800*, although more carbon is retained in the soils resulting from lower soil temperatures in *I-3000*. The relatively small responses of the terrestrial biosphere to the injections, compared to the *RCP 8.5 control run*, show a similar progression, although with

different amplitudes, as illustrated in Figure 4 f, e. After the injection period, this is especially reflected by the apparent synchronous increase in land carbon around the year 2600 and the synchronous decrease around the year 2770 (Fig. 4 e). This is a result of a slightly different phase of small variations in the total land carbon content of the *control run* (Figs. 4 g, S2 a, b), which is the only simulation that has not seen any atmospheric $CO_2$ reduction. However, due to the same amount of

atmospheric carbon being removed and injected into the ocean, the *With Emissions* runs have a similar climatic state throughout the simulations with comparable changes in global mean air and soil temperatures (between 0.1% to 0.3% less) and precipitation over land (between 0.1% to 0.4% more) when compared to the control run (Figs. 5 a, b, e). The high synchronicity (Fig. 4 e) can be further explained by the fact that in the *With Emissions* simulations the same biome regions are sensitive to the changes in temperature (Figs. 5 a, b), although the magnitudes of the absolute changes in land carbon differ between the injection runs (Figs. S3-S5). These regions are predominantly located at transition zones of different plant functional types that are in competition which each other and thus shift from one to another, leading to small changes in land carbon. The offset between *I-800* and *I-3000* (Fig. 4 e) is caused by higher soil respiration in *I-800* (Fig. 5 d), which is due to slightly higher global mean air and soil temperatures (Fig. 5 a, b).

For *I-1500*, an unexpected oceanic carbon uptake event is observed from the last year of the injection period (Figs. 4 c, d). This is caused by a large temporary carbon flux from the atmosphere into the ocean (Fig. 4 d), with a total of ~13 GtC taken up in a region of the Southern Ocean (~ 0°: 20°E; 60°: 70S°) between the years 2119 and 2209 (Fig. S6). Because this event is not simultaneously present in the reference simulation without injection, the difference in atmospheric carbon between run *I-1500* and the reference run even exceeds the globally injected carbon between the years 2189 and 2262 (Fig. 4 b). For standard accounting of carbon removed from the atmosphere with respect to a reference simulation, this would correspond to sequestration effectiveness greater than 100%. The oceanic *netFS* is just less than 100% of the GIC (Fig. 4 c). Our analysis for *I-1500* suggests that the regional carbon uptake is due to an intermittent ocean deep convection event that occurs in the *I-1500* simulation. Using an earlier version of the UVic model (version 2.8), Meissner et al. [2007] found that under a $CO_2$ concentration of 440 ppm or higher, the modeled climate system started oscillating between a state with open-ocean deep convection in the Southern Ocean, causing massive bottom water formation, and a state without. In their runs, which were spun up to equilibrium under constant atmospheric $CO_2$, the simulated deep convection event led to a rapid increase in atmospheric temperatures, carbon outgassing and a subsequent increase in atmospheric $CO_2$ concentrations. In contrast to Meissner et al. [2007], we here find that a deep convection event during a transient high $CO_2$ emission scenario can result in carbon uptake, as also found in CMIP5 model runs [Bernardello et al., 2014]. This can be explained by the fact that the $pCO_2$ of the old (pre-industrial) water masses that reach the surface during deep convection is lower than the

atmospheric $pCO_2$ in the *I-1500* simulation at the end of the 22$^{nd}$ century. Compared to the injected carbon content of 70 GtC at the end of the injection period, the deep convection event leads to a significant carbon uptake of about 19 %. Compared to the oceanic uptake of anthropogenic $CO_2$ by the end of the simulation, the carbon uptake associated with the deep convection event amounts to less than 1 %. The deep convection event also causes the ocean to lose a substantial amount of heat, which causes regional warming and thus partially counteracts the cooling effect associated with the direct $CO_2$ injection in I-1500. This is also reflected in a slower increase in total land carbon (Fig. 4 e, f) through more soil respiration than in I-800 and I-3000.

Recurring open ocean deep convection in the Southern Ocean has been found in many CMIP5 models (Lavergne et al., 2014) and also in the Kiel Climate Model, for which the driving mechanism could be linked to internal climate variability [Martin et al., 2013]. Although the modeled deep convection events feature similarities to processes associated with the Weddell Polyna of the 1970s [Martin et al., 2013], uncertainty remains regarding their realism. An important model constraint in this respect is a coarse grid resolution, which hinders, for instance, the correct representation of bottom water formation processes on the continental shelf and instead might favor open-ocean deep convection [Bernardello et al., 2014].

It is intriguing that among nineteen millennial-scale simulations performed for this study, a deep convection event occurred only in three simulations, the *I-1500*, an injection run with a ten year injection period (not shown) and the *Direct Air Capture run*. Apparently, small internal variability combined with certain $CO_2$ levels can give rise to such events [Meissner et al., 2007]. The only means to discriminate between the feedbacks of the ocean deep convection event, which are driven by the removal of atmospheric carbon and the little internal variability in the UVic model, would be to run ensembles with different initial conditions. This is how one would also discriminate between other feedbacks and internal variability in models with more intense - and more realistic - levels of internal variability. Furthermore, ensembles would allow one to assess of the robustness of the occurrence of ocean deep convection events, which might become more significant or different for slightly perturbed initial conditions. Such open-ocean deep convection can cause an inter-model spread in projections of future ocean carbon uptake [Bernardello et al., 2014] and may make accounting for the injected $CO_2$ as the net fraction stored (*netFS*) very difficult. As shown by the dashed lines in Figure 4, the fraction of the injected $CO_2$ retained (*FR*), that could in principle be tracked via a marker tracer, is more robust to internal variability of the model and,

presumably, of the real world. A pragmatic and robust way to account for the storage of injected $CO_2$ might therefore well be based on *FR* despite its neglect of carbon cycle and climate feedbacks. To account for these feedbacks, *FR* could possibly be augmented by some model-derived correction factors to account for the ensemble-averaged interaction of the ocean with the other carbon pools under changing climate conditions.

### 3.4.3 Sensitivity to variations in the $CO_2$ fertilization parameterization

Here we show how varying the $CO_2$ fertilization parameterization in the perturbed injection runs (i.e. i.e. *I-800$_{CO2\_fert\_high\ and\ low}$* and *I-3000$_{CO2\_fert\_high\ and\ low}$*) changes carbon cycling and the leakage of injected $CO_2$, when compared to the standard *I-800* and *I-3000* experiments of the *With Emissions* simulations.

As illustrated by the error bars in Figure 6 c, varying the $CO_2$ fertilization effect impacts the targeted atmospheric carbon reduction in *I-800* of the *With Emissions* experiments, leading to a difference of -0.5 GtC to 0.02 GtC in the year 2120 and of 0.4 GtC to 1.1 GtC in the year 3020. Absolute changes in total oceanic carbon are also rather insensitive in these simulations with differences of only about -0.7 GtC to 0.4 GtC (0.01 GtC to 0.3 GtC) in the year 2120 (3020) (Figs. 6 d, e). Accordingly, the difference in the net fraction stored (*netFS*) in *I-800* lies between -1% and 0.5% (Fig. 6 b) at the respective times. The slight differences in the fraction retained in I-800 (between -0.2 % and 0.3% at the respective times) are due to a slightly different climate in the perturbed simulations, when compared to the standard *With Emissions* runs, which is caused by the different atmospheric carbon concentrations (Fig. 6 c).

Absolute changes in terrestrial land carbon uptake and total land carbon show the largest sensitivities to the scaled $CO_2$ fertilization effect in *I-800* (Figs. 6 f, g). By the end of the injection period, the difference in total land carbon between *I-800* and the *RCP 8.5 control run*, shows that this terrestrial response could result in almost the same or less carbon storage, depending on the scaling of the $CO_2$ fertilization parameterization (Fig. 6 g). Higher $CO_2$ fertilization, i.e. *I-800$_{CO2\_fert\_high}$*, leads to a higher carbon flux from the atmosphere to land than in *I-800*, which counteracts the lower $CO_2$ fertilization effect that occurs in the standard *I-800* because of less atmospheric carbon, when compared to the *RCP 8.5 control run* [see section 3.4.1]. This results in more land carbon of about 1.1 GtC (Fig. 6 g). The opposite is true for *I-800$_{CO2\_fert\_low}$*, leading to less land carbon by about 0.4 GtC in the year 2120, when compared to the difference between *I-800* and the *RCP 8.5 control run*. By the end of the simulation, the perturbed injection simulation *I-800$_{CO2\_fert\_high}$* has about 0.4 GtC less land carbon, relative

to the difference of *I-800* and the *control run*, which is caused by a slightly stronger cooling effect, because there is less atmospheric carbon than in I-800 (Fig. 6 g). This cooling also results in less soil respiration. *I-800$_{CO2\_fert\_low}$* has about 1.3 GtC less land carbon at the end of the simulations, when compared to the absolute change between *I-800* and the respective control run. This can be explained by the reduced $CO_2$ fertilization effect that has led to a decreased NPP and consequently to a reduced soil respiration, when compared to *I-800*.

The magnitude of the responses that can be seen in the perturbed injection runs I-3000$_{CO2\_fert\_high}$ and I-3000$_{CO2\_fert\_low}$ are similar as in the perturbed *I-800* runs.

Although the above response is informative, the future strength of the $CO_2$ fertilization effect also depends on other factors, such as water and nutrient availability [IPCC, 2013], which may be poorly simulated by our model. A key update since the Fourth Assessment Report by the IPCC is the implementation of nutrient dynamics in some of the CMIP5 land carbon models, such as in the NORESM-ME and CESM1-BGC models [Arora et al., 2013; Hajima et al., 2014]. There is high confidence that low nitrogen availability will limit land carbon uptake. Models that combine nitrogen limitation with rising $CO_2$ as well as changes in temperature and precipitation, predict a larger increase in projected future atmospheric $CO_2$ for a given $CO_2$ emission scenario [e.g., IPCC, 2013, Hajima et al, 2014]. Models including terrestrial nutrient limitation would likely be subject to a smaller terrestrial response if direct $CO_2$ injections into the deep ocean occurred. Thus, the introduction of nitrogen limitation in the land component of the UVic model would presumably result in less total simulated land carbon, because of lower NPP and soil respiration throughout the simulation, when compared to the terrestrial response in the shallow injection run (I-800) or for delayed emissions.

## 4. Conclusions

We use an Earth System Model of intermediate complexity to simulate direct $CO_2$ injections into the deep ocean under a high $CO_2$ emission scenario. The model-predicted fractions retained (*FR*) are found to be within the range of the values found by Orr et al. [2001]. In agreement with earlier studies [Jain and Cao, 2005] we also find that the *FR* is enhanced as global warming progresses. In our simulations, this enhancement amounts to about 7% to 16% at the end of the simulations (year 3020). Injection sites in the Pacific are the most effective ones on the millennial time scale considered in our simulations. The neglect of the effect of the dissolution of $CaCO_3$ sediments near or downstream of the injection sites (see section 2.2)

may have led to an underestimation of the *FR* and *netFS* in our injection experiments. The impact of this process would

presumably be largest in the Atlantic due to the lower abundance of $CaCO_3$ sediments in the Pacific and Indian Ocean.

The response of the carbon cycle during and after the injections is dominated by the partial outgassing of injected $CO_2$ and a reduced rate of air-sea gas exchange compared to the control run without injection. Relative to the control run, the model's terrestrial ecosystems respond to the marine $CO_2$ injection and reduced atmospheric $CO_2$ concentrations via a reduced $CO_2$ fertilization effect and a temperature-related decrease in soil respiration. This leads to a maximum reduction in

total land carbon by about 4 GtC (relative to the control run) during the injection period in all *With Emissions* simulations (Fig. 4 e). After the injection period, total land carbon becomes higher than in the control simulation, mainly due to a terrestrial carbon cycle-climate feedback, with a maximum increase of about 5 GtC for *I-3000* in the year 2230 (Fig. 4 e).

Further, we find that varying the $CO_2$ fertilization parameterization results in changes of the targeted atmospheric carbon reduction in *I-800* and *I-3000* of the *With Emissions* simulations that lay between 0.2% and 2% less atmospheric

carbon at the end of the injection period (year 2120) and between 9% less and 1% more atmospheric carbon at the end of the simulations. The sensitivity of the terrestrial carbon cycle to the different $CO_2$ fertilization parameterizations in *I-800* and *I-3000* of the *With Emissions* runs ranges from 30% less to 98% more land carbon by the year 2120 and up to 108% less land carbon by the end of the simulations. The larger signal of the terrestrial response to the scaled $CO_2$ fertilization parameterization, when compared to the targeted atmospheric carbon reduction, highlights that further research on the future

strength of terrestrial carbon cycle feedbacks is needed if direct $CO_2$ injections were to be seriously considered.

Furthermore, the influence of the highly uncertain carbon-cycle and climate feedbacks in our findings, in addition to the sporadic deep convection event in *I-1500*, illustrates the difficulty of quantitatively detecting, attributing, and eventually accounting for, carbon storage and carbon fluxes generated by individual carbon sequestration measures even in relatively coarse-resolution models with little internal climate variability ("noise"). Nevertheless, our findings point to the importance

of accounting for all carbon fluxes in the carbon cycle and not only for those of the manipulated reservoir, to obtain a comprehensive assessment of direct oceanic $CO_2$ injection in particular and carbon sequestration in general.

**Acknowledgments**

The model data used to generate the table and figures will be available at http://thredds.geomar.de/thredds/catalog-opene-access.html. The Deutsche Forschungsgemeinschaft (DFG) financially supported this study via the Priority Program 1689. We thank Torge Martin, Wolfgang Koeve, Nadine Mengis, Julia Getzlaff, Levin Nickelsen, Peter Vandromme, Markus Pahlow, Wilfried Rickels and Ell Yuming Feng for their thoughtful discussions and advice.

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

**Table 1:** Overview of all conducted simulations and their anthropogenic forcing. The 'X' denotes that the respective forcing is applied.

| Simulation | RCP 8.5 $CO_2$ emission scenario from 2006 to | | Extended RCP 8.5 $CO_2$ emission scenario from 2100 until 2500 | Constant $CO_2$ emissions of 1.48 GtC yr$^{-1}$ from 2500 onwards | Continuous $CO_2$ injections into deep ocean of 0.7 GtC yr$^{-1}$ from 2020 to 2120 | 0.7 GtC y$^{-1}$ continuously subtracted from $CO_2$ emissions from 2020 to 2120 |
|---|---|---|---|---|---|---|
| | 2020 | 2100 | | | | |
| RCP 8.5 control run of With Emissions (WE) simulations | | X | X | X | | |
| I-800 WE | | X | X | X | X | X |
| I-1500 WE | | X | X | X | X | X |
| I-3000 WE | | X | X | X | X | X |
| RCP 8.5 control run of Complete Mitigation (CM) simulations[1] | X | | | | | |
| I-800 CM | X | | | | X | X |
| I-1500 CM | X | | | | X | X |
| I-3000 CM | X | | | | X | X |
| Direct Air capture run | | X | X | X | | X |
| RCP 8.5 control$_{Co2\_fert\_high}$ | | X | X | X | | |
| I-800$_{Co2\_fert\_high}$ | | X | X | X | X | X |
| I-3000$_{Co2\_fert\_high}$ | | X | X | X | X | X |
| RCP 8.5 control$_{Co2\_fert\_low}$ | | X | X | X | | |
| I-800$_{Co2\_fert\_low}$ | | X | X | X | X | X |
| I-3000$_{Co2\_fert\_low}$ | | X | X | X | X | X |

[1]After the year 2020, CM simulations continue without $CO_2$ emissions until 3020.

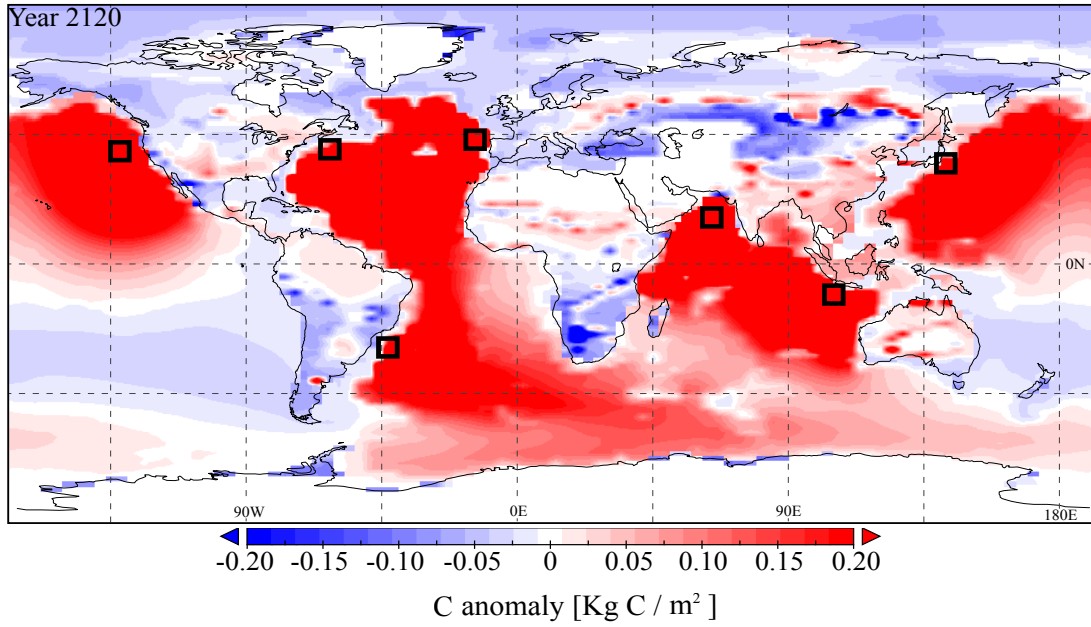

**Figure 1:** Absolute changes in oceanic and land carbon between I-3000 and the RCP 8.5 control run (I-3000 simulation minus RCP 8.5 control run) at the end of the injection period (year 2120). The black rectangles represent the locations of the seven injection sites, where the injections occurred in the center of the black rectangles.

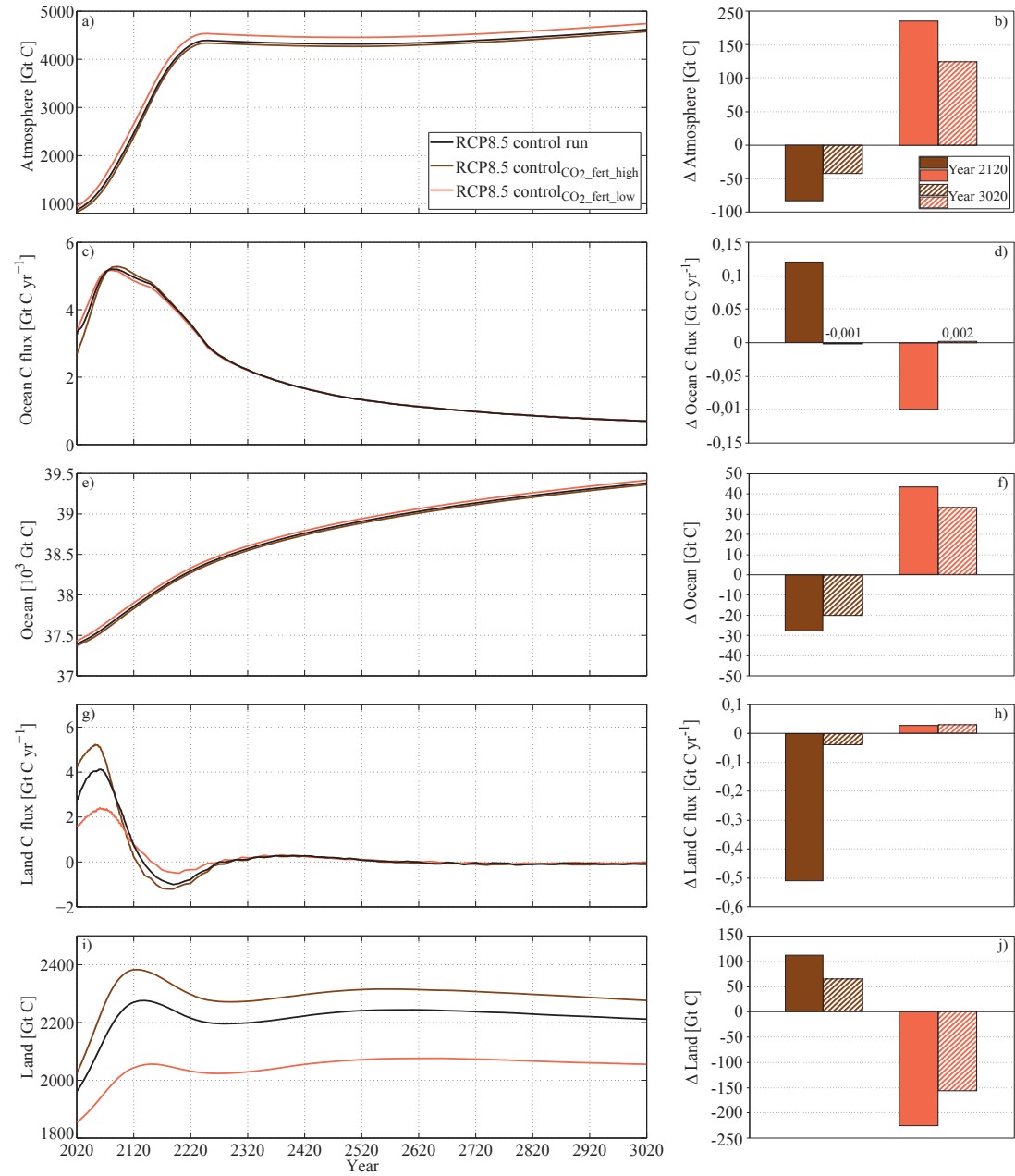

**Figure 2:** Globally integrated carbon of the RCP 8.5 control run, the RCP 8.5 control$_{CO2\_fert\_high}$ and RCP 8.5 control$_{CO2\_fert\_low}$ for (a) total atmospheric carbon, (c) carbon flux from atmosphere to ocean, (e) total oceanic carbon, (g) carbon flux from atmosphere to land, and (i) total land carbon. Difference in carbon between the RCP 8.5 control$_{CO2\_fert\_high}$ (brown) (RCP 8.5 control$_{CO2\_fert\_low}$, orange) and the RCP 8.5 control run (perturbed control runs minus RCP 8.5 control run) for the years 2120 (filled) and 3020 (hashed) for (b) globally integrated total atmospheric carbon (d) globally integrated carbon flux from atmosphere to ocean (f) globally integrated total oceanic carbon (h) globally integrated carbon flux from atmosphere to land, and (j) globally integrated total land carbon.

**Table 2:** Comparison of fractions retained (*FR*) between Orr et al. [2001; Orr, 2004] (Full Range of their Global Efficiency, which is the same as the *FR* defined in section 2.2 and is based on seven OGCM and one zonally averaged model results) and our Complete Mitigation (CM) and With Emissions (WE) simulations for all injection sites (Global) and on an inter-basin level for the Atlantic sites (Bay of Biscay, New York, Rio de Janeiro), the Pacific sites (San Francisco, Tokyo) and the Indian sites (Jakarta, Mumbai). The *FR* values [%] are given for the last year of the injections (2119), 500 years after the simulations started (2519) and for the last year of the simulations (3019). For each entry of the table, numbers to the left of the vertical bar denote results of the CM runs, numbers to the right results of the WE runs. Note that the illustrated years refer to our simulations, ranging from year 2020 until the year 3020. The GOSAC-OCMIP simulations started in the year 2000 and ended in the year 2500 [Orr et al., 2001].

| Overview of *FR* [%] | I-800 | | | I-1500 | | | I-3000 | | |
|---|---|---|---|---|---|---|---|---|---|
| | Year | | | Year | | | Year | | |
| | **2119** | **2519** | **3019** | **2119** | **2519** | **3019** | **2119** | **2519** | **3019** |
| **Full Range** [Orr et al., 2001; Orr, 2004] | 65 - 84 | 15 - 38 | - | 81 - 96 | 32 - 57 | - | 97 - 100 | 49 - 93 | - |
| *CM \| WE* **Global** | 68 \| 75 | 17 \| 30 | 8 \| 17 | 92 \| 95 | 40 \| 56 | 20 \| 35 | 99 \| 100 | 65 \| 76 | 38 \| 54 |
| *CM \| WE* **Atlantic sites** (70°N:35°S) | 53 \| 64 | 9 \| 20 | 5 \| 11 | 85 \| 91 | 30 \| 46 | 16 \| 28 | 97 \| 99 | 62 \| 75 | 37 \| 54 |
| *CM \| WE* **Pacific sites** (65°N:35°S) | 78 \| 81 | 27 \| 45 | 13 \| 29 | 97 \| 98 | 61 \| 77 | 34 \| 55 | 99 \| 100 | 86 \| 93 | 59 \| 75 |
| *CM \| WE* **Indian sites** (20°N:35°S) | 80 \| 84 | 17 \| 29 | 6 \| 14 | 96 \| 97 | 34 \| 49 | 13 \| 25 | 99 \| 100 | 50 \| 65 | 20 \| 34 |

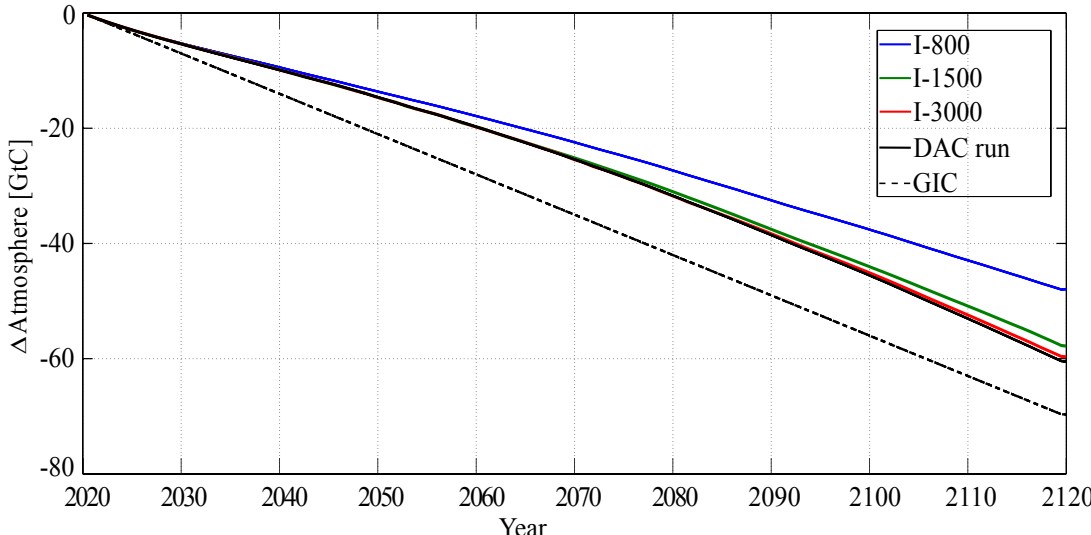

**Figure 3:** Absolute change in atmospheric carbon in the Direct Air Capture run (DAC) and in the With Emissions simulations, relative to the RCP8.5 control run. The black dashed line denotes the globally injected carbon (GIC), which is subtracted from the emission forcing (see section 2.2).

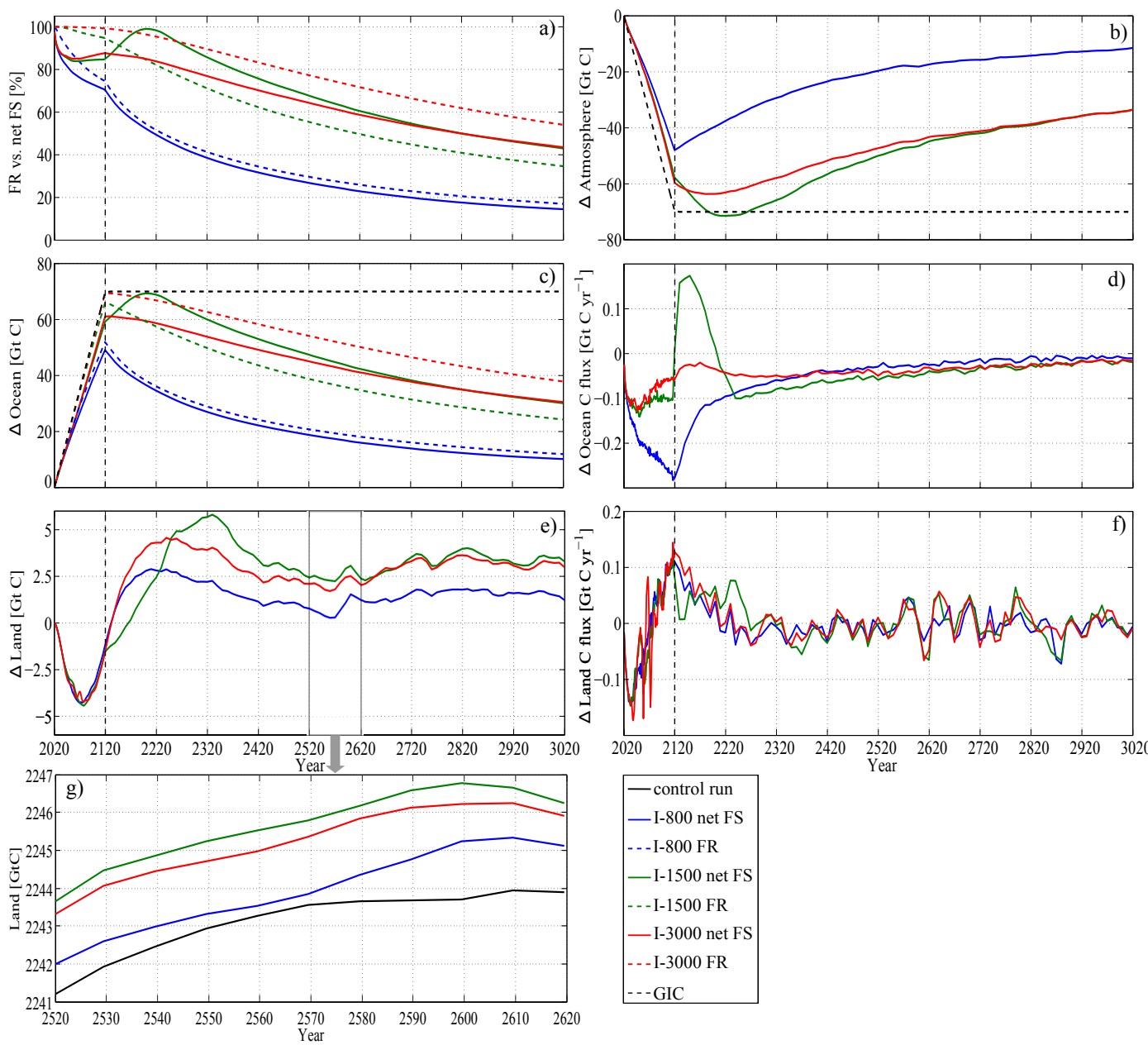

**Figure 4:** (a) Comparison of the fractions retained (FR, dashed) and the net fractions stored (netFS, solid) of the With Emissions (WE) simulations. Absolute changes in carbon between the WE simulations and the RCP 8.5 control run (WE simulations minus RCP 8.5 control run) for (b) globally integrated total atmospheric carbon, (c) globally integrated total oceanic carbon, (d) globally integrated carbon flux from atmosphere to ocean, (e) globally integrated total land carbon, (f) globally integrated carbon flux from atmosphere to land, and (g) absolute values of globally integrated total land carbon of the WE simulations and the RCP 8.5 control run from year 2520 to 2620. The globally injected carbon is denoted as GIC. The vertical dashed black lines indicate the end of the injection period.

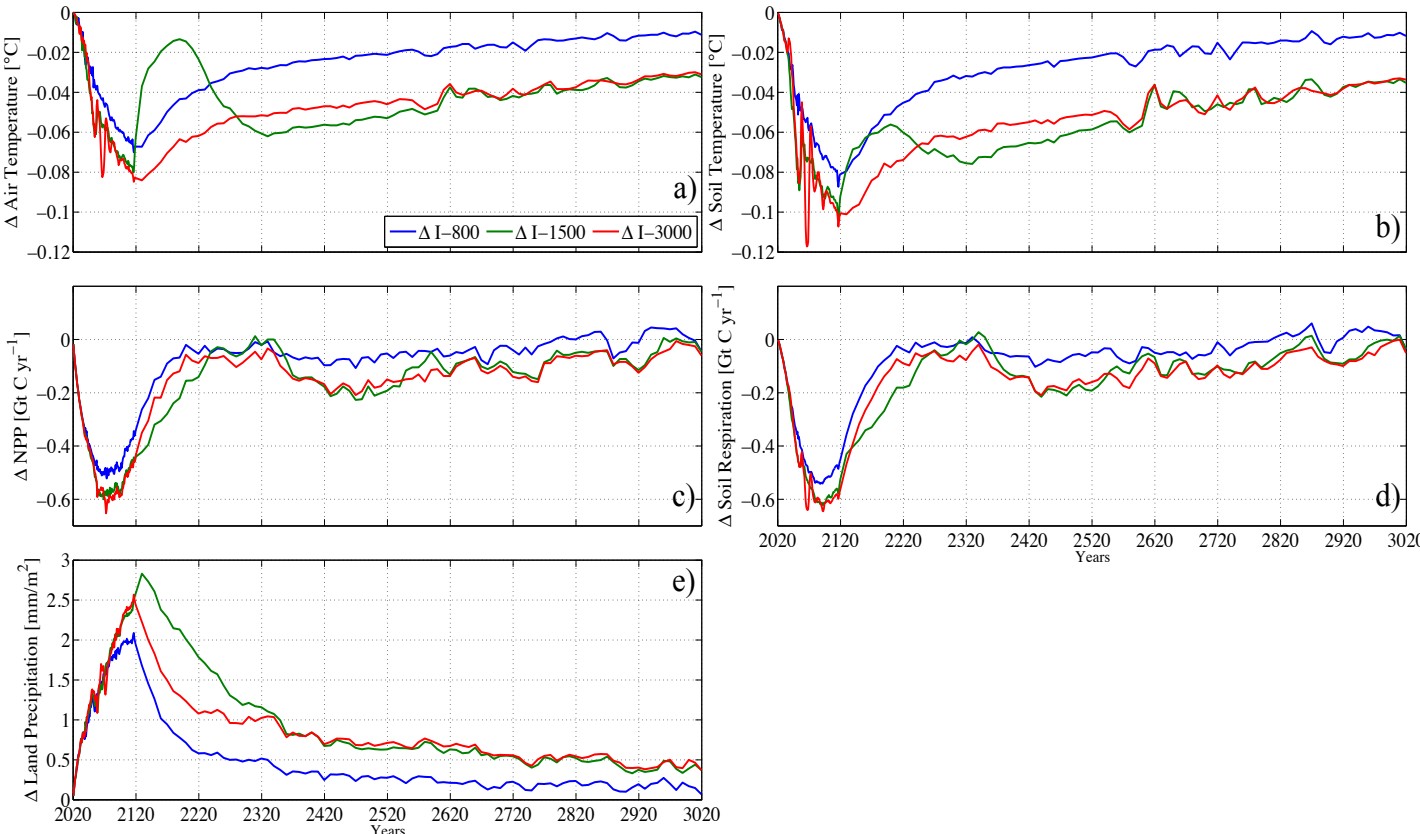

**Figure 5:** Absolute changes between the WE simulations and the RCP 8.5 control run for (a) global mean surface air temperature, (b) global mean soil temperature, (c) globally integrated net primary productivity on land, (d) globally integrated soil respiration, and (e) global mean precipitation over land.

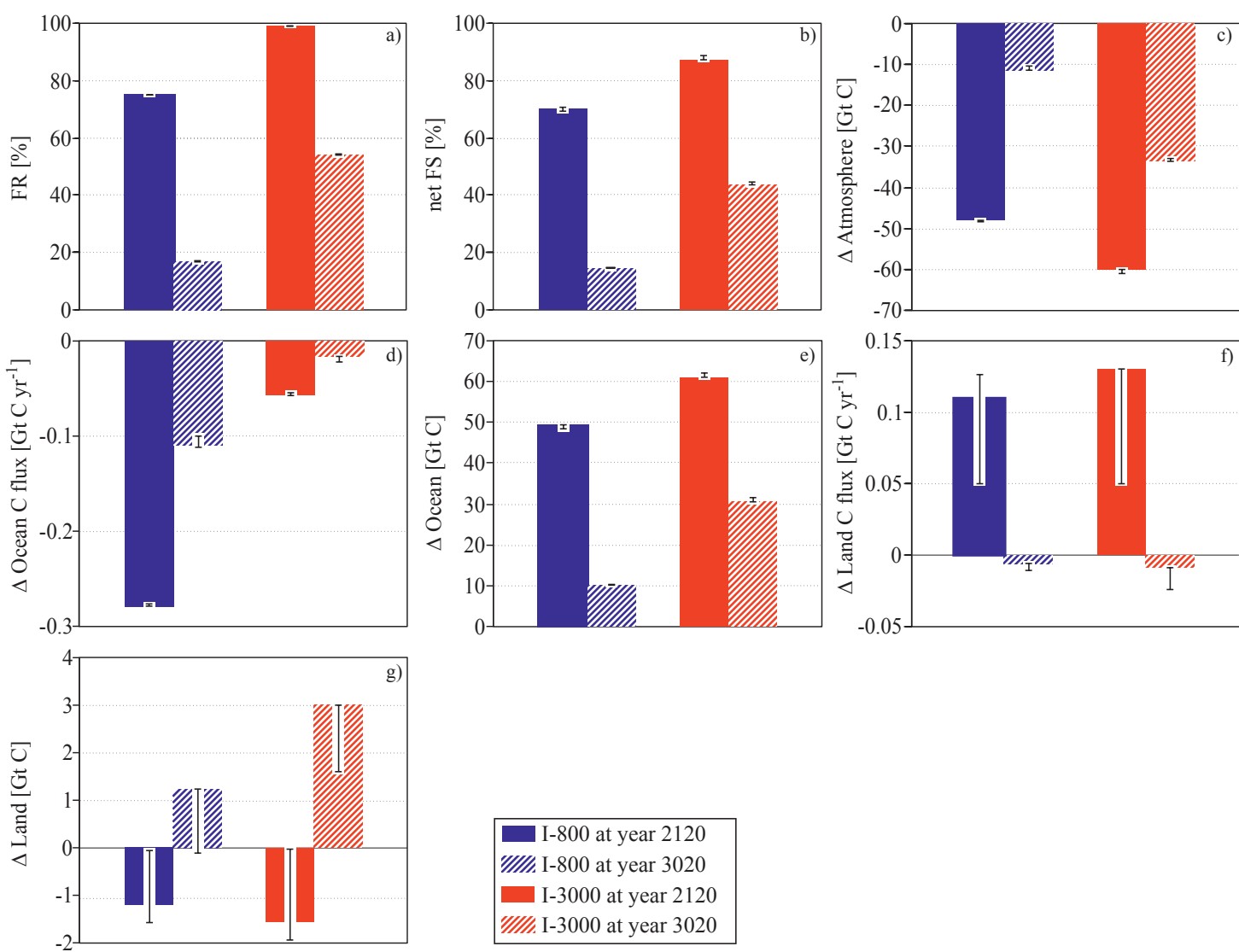

**Figure 6:** (a) Fraction retained (FR) for I-800 (blue) and I-3000 (red) for the years 2120 (filled) and 3020 (hashed) with error bars. The error bars are in all panels defined as the difference of absolute changes between the perturbed injection runs and the respective control runs and the absolute change between the unperturbed injection runs and the control run of the With Emissions simulations. (b) Net fraction stored (netFS) for I-800 and I-3000 for the years 2120 and 3020 with error bars. Absolute changes in carbon between I-800 (I-3000) and the RCP 8.5 control run (With Emissions simulations minus RCP 8.5 control run) error bars for the years 2120 and 3020 for (c) globally integrated total atmospheric carbon, (d) globally integrated carbon flux from atmosphere to ocean, (e) globally integrated total oceanic carbon, (f) globally integrated carbon flux from atmosphere to land, and (g) globally integrated total land carbon.