# Peer review of "Revisiting ocean carbon sequestration by direct injection: A global carbon budget perspective"

_Earth System Dynamics, 2016_

## Referee Comment (RC1) · Anonymous Referee #1 · 27 May 2016

The authors investigate the impacts of ocean carbon injection (and of direct carbon capture and storage with no leakage) on the carbon inventories of the atmosphere, the ocean, and the land biosphere using the UVIC model. This is a solid study that should be published after taking into account the following comments:

1) The authors evaluate the impact of climate change on the fraction retained by comparing their complete mitigation (CM) simulations without emission forcing after 2020 and the RCP8.5 simulations with continued emissions (WE) (Line 181). They conclude (line 182) that larger climate change in RCP8.5 leads to a higher fraction of injected carbon retained in the ocean (FR).

I doubt that the difference between the CM and RCP85-WE simulations is indicative of climate change. I suspect that the higher fraction retained in the CM compared to the

WE simulation is largely the result of differences in the Revelle factor/carbonate chemistry. The higher carbon emissions under RCP8.5 lead to a higher atmospheric and oceanic CO2 and a higher Revelle factor. In turn a smaller fraction of anthropogenic carbon ends up in the ocean in the RCP8.5 case compared to the zero emission CM case. As in the long run, both simulations with and without ocean injection tend to achieve the same carbon partitioning between the ocean and the atmosphere (when neglecting ocean-sediment and weathering fluxes as done here) this mechanisms also affects the fraction retained. More injected carbon remains in the ocean for the low than for the high emission case.

A proper evaluation of the climatic impacts would require an RCP8.5 simulations with carbon emissions, but with radiative forcing from anthropogenic agents set to zero. Then, climate would remain at equilibrium while atm. CO2 and carbonate chemistry would still change.

(Alternatively, I may misunderstand the experimental protocol. This would then require a clarification in the method section.)

2) A caveat of this study is that ocean sediments and the effect of calcium carbonate dissolution (also known as calcium carbonate compensation) is not considered. This caveat should be addressed in the introduction and conclusion section. This mechanisms could be relatively important as ocean carbon injection may bring the excess carbon close to deposits of calcium carbonate and thus would permit carbonate dissolution to occur on much faster time scale than for emissions into the atmosphere.

3) The marker tracer used to compute the fraction retained should be explained in detail in the method section. As the fraction retained (FR) is a central metric in this study, it is not enough to refer to the literature.

Further comments

Line 44: "reach a chemical equilibrium (mainly an equilibrium between the ocean and

atmospheric carbon reservoirs)." This statement is not completely true as carbonate compensation and weathering feedbacks are important for time scales longer than ∼5000 years.

L 93: What about non-CO2 forcings?

Line 127: could you please say a few more words about the diagnostic marker tracer. How is carbonate chemistry and air-sea and air-land flux computed for this tracer?

Line 183: I doubt that the FR remains higher with than without climate change. I also doubt that this statement applies to all time scales (longer than the simulations).

———————————————

---

## Referee Comment (RC2) · C. Heinze (Referee) · 30 May 2016

C. Heinze (Referee)

christoph.heinze@gfi.uib.no

The manuscript investigates the effect of direct oceanic water column CO2 injection on the redistribution of carbon under a high emission scenario following RCP8.5 its extension to 2300/2500 according to Meinshausen et al. (2011) and keeping emissions at a constant value until year 3020. The authors employ an Earth system model of intermediate complexity (UVic EMIC) and a standard protocol for prescribing the CO2 injections. The study goes beyond the state-of-the-art by confronting not only an ocean biogeochemical model (with atmospheric reservoir) but a coupled Earth system model including also a terrestrial biosphere component (and a simple atmosphere representation) with ocean CO2 injections. The model runs are carried out in a technically correct way as far as one can judge from the description. If I am not mistaken, the main result of the study is the following: CO2 injection does not change the control run result for

land carbon storage in a significant way for the forcing and injection protocol as applied. The last sentence in the conclusions (l. 348-350) maybe true in general but is hardly backed up by this particular study. The CMIP5 inter-model spread in land carbon storage change is much larger at year 2100 (Jones et al., J.Clim., 2013) than the amount discussed here as caused by ocean injection of $CO_2$. The manuscript confirms previous studies: A part of the injected $CO_2$ will outgas at a certain point in time, leading to less than 100% efficiency of the injection with respect to keeping anthropogenic excess $CO_2$ isolated from the atmosphere.

The authors correctly motivate their study with the current discussion on feasible mitigation targets to limit radiative warming to 2deg or 1.5deg C with respect to the pre-industrial. Respective emission scenarios would require at some point negative emissions. Why did the authors choose the business as usual strong warming scenario for their study? The amount of injected $CO_2$ is small in view if the $CO_2$ emissions in the RCP8.5 emission driven case. A more modest emission scenario would have been may be more appropriate in view of the amount of injected $CO_2$ as used here.

The terrestrial carbon cycle model used here is originally based on TRIFFID. This model has at times shown a more sensitive behaviour to forcing than other models (see e.g. Friedlingstein et al., J. Clim., 2006/C4MIP, where both the Hadley Centre model and the UVIC model show significant outgassing after 2050). Would results with other terrestrial modules potentially show an even smaller deviation from the control run for the injection scenarios? The spread among different terrestrial carbon cycle modules concerning $CO_2$ uptake in Earth system models is large, also in view of the effect of nitrogen cycle perturbations. The fluxes as presented in the paper should have been discussed in view of also these uncertainties. The authors correctly mention the as yet difficult to quantify $CO_2$ fertilisation effect on land as large source of uncertainty.

The authors say that "direct injection of $CO_2$ is presently in conflict with ..." international protocols/conventions. This is correct but may also be an understatement. Direct $CO_2$ injection has been abandoned as a mitigation option because its environmental

risks are potentially large (see WBGU report, 2006, for a summary of related risks, http://www.wbgu.de/en/special-reports/sr-2006-the-future-oceans/). The injection protocol of OCMIP/GOSAC as applied in the study does not account for the potential of fast rising bubbles after $CO_2$ injection (e.g., Bigalke et al., Environ. Sci. Technol., 2008). Deeper ocean environments are sensitive to small pH variations (e.g., Gehlen et al., Biogeosciences, 2014). These aspects should be discussed in order to avoid misunderstandings by non-expert readers.

The authors discuss a transient Southern Ocean fluctuation of their model on one hand, and the lack of realistic internal variability in the EMIC employed on the other hand. The strength of EMICs is their low demand for computational resources. They would be suited to carry out ensemble simulations with large numbers of members. This advantage could have been used to assess the robustness of the results. Maybe these would have become more significant or different for slightly perturbed initial conditions in an ensemble simulation?

Deep injection of $CO_2$ could potentially accelerate neutralising fossil fuel $CO_2$ by dissolution of $CaCO_3$ from the sea-floor. Usually, on a 1000-years-time scale, the negative carbon cycle feedback through $CaCO_3$ sediment dissolution is not important but rather on a several 10,000 year time scale (Archer, J.Geophys.Res., 2005). Water column injection potentially could change this, though injection in the deep Pacific, where injection would be most effective, $CaCO_3$ sediment is scarce. Nevertheless this aspect would warrant discussion. Is the (presumably small) $CaCO_3$ effect larger than the land biosphere effect as discussed here?

A successful revision of this honest study would hopefully make the results more significant in quantitative terms.

Small details:

Abstract, l. 17: An . . . feature are effects (conflict singular/plural)

I find the introduction of the acronyms CM, WE, DAC, and GIC not helpful. One can spell the terms out (maybe in italics).

l. 136: misplaced comma

l. 183: comma after simulations required

Figure 1: The small rectangles with injection sites are difficult to identify.

Figure S2 should be placed in the main section. It shows the small effects.

I do not want to stay anonymous.

---

## Author Comment (AC4) · 26 Aug 2016

***Responses in Italic***

*First of all, the authors thank Prof. Christoph Heinze very much for his thoughtful and constructive comments and advice.*

The manuscript investigates the effect of direct oceanic water column $CO_2$ injection on the redistribution of carbon under a high emission scenario following RCP8.5 its extension to 2300/2500 according to Meinshausen et al. (2011) and keeping emissions at a constant value until year 3020. The authors employ an Earth system model of intermediate complexity (UVic EMIC) and a standard protocol for prescribing the $CO_2$ injections. The study goes beyond the state-of-the-art by confronting not only an ocean biogeochemical model (with atmospheric reservoir) but a coupled Earth system model including also a terrestrial biosphere component (and a simple atmosphere representation) with ocean $CO_2$ injections. The model runs are carried out in a technically correct way as far as one can judge from the description. If I am not mistaken, the main result of the study is the following: $CO_2$ injection does not change the control run result for land carbon storage in a significant way for the forcing and injection protocol as applied. The last sentence in the conclusions (l. 348-350) maybe true in general but is hardly backed up by this particular study. The CMIP5 inter-model spread in land carbon storage change is much larger at year 2100 (Jones et al., J.Clim., 2013) than the amount discussed here as caused by ocean injection of $CO_2$. The manuscript confirms previous studies: A part of the injected $CO_2$ will outgas at a certain point in time, leading to less than 100% efficiency of the injection with respect to keeping anthropogenic excess $CO_2$ isolated from the atmosphere.

*Yes, we agree with the reviewer that the universality of the last concluding sentence is not completely backed up by our study. This would have required the comparison of the injection simulations with and without the land module. We will rephrase the last sentence of the conclusion, accordingly.*

*The comment related to the CMIP5 inter-model spread in land carbon storage change is discussed below.*

The authors correctly motivate their study with the current discussion on feasible mitigation targets to limit radiative warming to 2deg or 1.5deg C with respect to the pre-industrial. Respective emission scenarios would require at some point negative emissions. Why did the authors choose the business as usual strong warming scenario for their study? The amount of injected $CO_2$ is small in view if the $CO_2$ emissions in the RCP8.5 emission driven case. A more modest emission scenario would have been maybe more appropriate in view of the amount of injected $CO_2$ as used here.

*Our choice of the experimental design is motivated by the current trend of $CO_2$ emissions, which continues to follow largely the trajectory of the RCP 8.5 emission scenario [Peters et al., 2013] and also by our choice of the objective of our study, i.e., to investigate the response of the global carbon cycle during and after the direct $CO_2$ injections, considering a strong perturbation of the climate system. This has helped to investigate the effect of climate-induced changes on the fraction retained by comparing our 'Complete Mitigation runs' with the 'With Emissions simulations' (section 3.3). The justification of the small injection rate is that we wanted to compare and validate the fraction retained as well as the changes in seawater chemistry to the results of Orr et al. [2001; Orr, 2004].*

The terrestrial carbon cycle model used here is originally based on TRIFFID. This model has at times shown a more sensitive behavior to forcing than other models (see e.g. Friedlingstein et al., J. Clim., 2006/C4MIP, where both the Hadley Centre model and the UVIC model show significant outgassing after 2050). Would results with other terrestrial modules potentially show an even smaller deviation from the control run for the injection scenarios? The spread among different terrestrial carbon cycle modules concerning $CO_2$ uptake in Earth system models is large, also in view of the effect of nitrogen cycle perturbations. The fluxes as presented in the paper should have been discussed in view of also these uncertainties. The authors correctly mention the as yet difficult to quantify $CO_2$ fertilization effect on land as large source of uncertainty.

*Yes, the authors agree that it is necessary to address and discuss the uncertainties related to the response of the terrestrial carbon cycle model to the direct $CO_2$ injections.*

*The process of $CO_2$ fertilization, which is here one of the dominant terrestrial carbon cycle feedbacks after $CO_2$ is injected, has direct relevance for the future trajectory of atmospheric $CO_2$ [IPCC, 2013] and thus for our targeted atmospheric carbon reduction of 70 GtC by the year 2120. The future strength of $CO_2$ fertilization in response to continued carbon emissions as in the 'With Emissions runs' is subject to the choice of the $CO_2$-fertilization parameterization and hence uncertain. In the new manuscript version we will provide a quantification of this uncertainty concerning the targeted atmospheric carbon reduction through direct $CO_2$ injections based on additional model runs, following the approach of Matthews [2007]. For these runs, we scaled the $CO_2$ sensitivity of the terrestrial photosynthesis model and have performed simulations of the RCP 8.5 control run, I-800 and I-3000, in which we have varied the strength of the $CO_2$ fertilization effect by increasing and decreasing it by 50% ($CO_2$ fert. =+50% / -50%) relative to the default model. We add a description of these simulations to the experimental design section.*

*In the results and discussion section (section 3.1), we describe carbon budgets of the perturbed control runs (RCP 8.5 $CO_2$ fert.=+50% and RCP 8.5 $CO_2$ fert.=-50%) and how these differ from the unperturbed control run. In addition, we illustrate the results in a new Figure 2, in which, in addition to time series of all control runs, we also show bar diagrams of the absolute changes in the carbon reservoirs and fluxes between the perturbed control simulations and the unperturbed control run for the years 2120 and 3020.*

*Further, in a new section (3.4.3), we show how the carbon budgets of the perturbed injections runs (I-800 $CO_2$ fert.=+50% / -50%, I-3000 $CO_2$ fert.=+50% / -50%), when compared to the respective control runs, differ from the anomalies of the injection runs of our original 'With Emissions simulations'. We further present the range of uncertainty for each carbon reservoir and flux at the year 2120 and 3020 in a new figure (new Fig. 6). For that purpose, we define the range of uncertainty as the difference of the absolute changes in atmospheric carbon between I-800 $CO_2$ fert.=+50% / -50% and the respective*

*control runs and the absolute change in atmospheric carbon between I-800 and the control run of the 'With Emissions simulations'.*

*Finally, we discuss the terrestrial response to injections in the un- and perturbed runs in the context of the large uncertainty range related to the inter-model spread in future land carbon storage change [e.g., Friedlingstein et al., 2006; Arora et al., 2013; Jones et al., 2013; Zickfeldt et al., 2013; Hajima et al., 2014]. We particularly discuss this in relation to the issue of nutrient limitation of photosynthesis currently missing in many terrestrial carbon cycle modules. There is high confidence that low nitrogen availability will limit land carbon uptake. Models that combine nitrogen limitation with rising $CO_2$ as well as changes in temperature and precipitation, predict a larger increase in projected future atmospheric $CO_2$ for a given $CO_2$ emission scenario [IPCC, 2013]. Models including terrestrial nutrient limitation are likely subject to a smaller terrestrial response to direct $CO_2$ injections into the deep ocean.*

The authors say that direct injection of $CO_2$ is presently in conflict with . . . international protocols/conventions. This is correct but may also be an understatement. Direct $CO_2$ injection has been abandoned as a mitigation option because its environmental risks are potentially large (see WBGU report, 2006, for a summary of related risks, http://www.wbgu.de/en/special-reports/sr-2006-the-future-oceans/). The injection protocol of OCMIP/GOSAC as applied in the study does not account for the potential of fast rising bubbles after $CO_2$ injection (e.g., Bigalke et al., Environ. Sci. Technol.,

2008). Deeper ocean environments are sensitive to small pH variations (e.g., Gehlen et al., Biogeosciences, 2014). These aspects should be discussed in order to avoid misunderstandings by non-expert readers.

*This is a very good point. We did not intend to trivialize the potential ecological risks of direct $CO_2$ injection into the deep ocean. We add a paragraph in the revised introduction section that addresses this issue. Further, we add the neglection of fast rising $CO_2$ bubbles [IPCC, 2005; Bigalke et al. 2008] in the experimental design section.*

The authors discuss a transient Southern Ocean fluctuation of their model on one hand, and the lack of realistic internal variability in the EMIC employed on the other hand. The strength of EMICs is their low demand for computational resources. They would be suited to carry out ensemble simulations with large numbers of members. This advantage could have been used to assess the robustness of the results. Maybe these would have become more significant or different for slightly perturbed initial conditions in an ensemble simulation?

*We have discussed possibilities to discriminate the impact of the natural variability (the deep convection) from the impact of $CO_2$ injections, for instance, during the injection phase before the onset of deep convection, or based on curve fitting of results from the other experiments, which show no deep convection events in the Southern Ocean. We came to the conclusion that no correct answer can be given without an ensemble simulation. Although the authors agree that it would be interesting and useful to perform an ensemble simulation with different initial conditions in order to assess the robustness of the ocean deep convection events, we feel that further analysis of it is beyond the scope of this study, which focuses on the response of the global carbon cycle during and after the $CO_2$ injections. In the manuscript we thus prefer to address this issue as done in line 420, but add a short discussion on the advantage of an ensemble simulation with respect to the reviewers comment.*

Deep injection of $CO_2$ could potentially accelerate neutralizing fossil fuel $CO_2$ by dissolution of CaCO3 from the sea floor. Usually, on a 1000-years-time scale, the negative carbon cycle feedback through CaCO3 sediment dissolution is not important but rather on a several 10,000 year time scale (Archer, J.Geophys.Res., 2005). Water column injection potentially could change this though injection in the deep Pacific, where injection would be most effective, CaCO3 sediment is scarce. Nevertheless this aspect would warrant discussion. Is the (presumably small) CaCO3 effect larger than the land biosphere effect discussed here?

*Yes, this is a very important point that we will discuss in the new results and discussion section (3.4.3). Dissolution of $CaCO_3$ sediments near or downstream of an injection site is expected to reduce*

*outgassing and increase the residence time of the injected $CO_2$. We expect a larger impact of this process in the Atlantic due to the low abundance of $CaCO_3$ sediments in the Pacific and Indian Ocean [Archer et al. 1998]. Model simulations by Archer et al. [1998] have shown that CaCO3 dissolution is sensitive to direct $CO_2$ injections throughout the Atlantic, but has led to only a slight impact on atmospheric $pCO_2$. A slightly modified trajectory of atmospheric $CO_2$ may further impact the terrestrial carbon pool and fluxes, resulting in a different terrestrial response as discussed in our manuscript. A quantitative answer, however, on how the marine CaCO3 sediments feedback or that of the land biosphere to direct $CO_2$ injections compare to each other can only be given by running the model with and without a sediment sub-model. This has not been done yet, mainly due to the several month long model runtime of the required spin-up of several 10,000 yrs. (Note that the code of UVic runs only on one processor, which typically simulates 200 model yrs. / day run time). We clarify in the experimental design and conclusion sections that we do not consider the effect of calcium carbonate sediments feedbacks in our direct $CO_2$ injection experiments.*

**With respect to small details:**

Abstract, l. 17: An . . . feature are effects (conflict singular/plural)

*Thank you for your careful reading. We correct this mistake.*

I find the introduction of the acronyms CM, WE, DAC and GIC not helpful. One can spell the terms out (maybe in italics).

*Yes, we agree that this could be confusing. We spell these acronyms out in italics.*

l. 136: misplaced comma

*Thank you, we correct this mistake.*

l. 183: comma after simulations required

*Thank you, we correct this mistake.*

Figure 1: The small rectangles with injection sites are difficult to identify.

*Yes, we thicken the black rectangles in Figure 1 to make them easier to identify.*

Figure S2 should be placed in the main section. It shows the small effects. I do not want to stay anonymous.

*Yes, we include Figure S2 in the main text as Figure 5.*

***References:***

*Archer, D., Khesghi, H., Maier-Reimer, E.: Dynamics of fossil fuel CO2 neutralization by marine CaCO3. Global Biogeochem. Cycles, 12259276, 1998.*

*Arora, V. K., Boer, G. J., Friedlingstein, P., Eby, M., Jones, C. D., Christian, J. R., Bonan, G., Bopp, L., Brovkin, V., Cadule, P., Hajima, T., Ilyina, T., Lindsay, K., Tjiputra, J. F. and Wu, T.: Carbon–Concentration and Carbon–Climate Feedbacks in CMIP5 Earth System Models, J. Clim., 26(15), 5289–5314, doi:10.1175/JCLI-D-12-00494.1, 2013.*

*Bigalke, N. K., Rehder, G. and Gust, G.: Experimental investigation of the rising behavior of $CO_2$ droplets in seawater under hydrate-forming conditions, Environ. Sci. Technol., 42(14), 5241–5246, doi:10.1021/es800228j, 2008.*

*Friedlingstein, P., Cox, P., Betts, R., Bopp, L., von Bloh, W., Brovkin, V., Cadule, P., Doney, S., Eby, M., Fung, I., Bala, G., John, J., Jones, C., Joos, F., Kato, T., Kawamiya, M., Knorr, W., Lindsay, K., Matthews, H. D., Raddatz, T., Rayner, P., Reick, C., Roeckner, E., Schnitzler, K.-G., Schnur, R., Strassmann, K., Weaver, A. J., Yoshikawa, C. and Zeng, N.: Climate–Carbon Cycle Feedback Analysis: Results from the C 4 MIP Model Intercomparison, J. Clim., 19(14), 3337–3353, doi:10.1175/JCLI3800.1, 2006.*

*Hajima, T., Tachiiri, K., Ito, A. and Kawamiya, M.: Uncertainty of concentration-terrestrial carbon feedback in earth system models, J. Clim., 27(9), 3425–3445, doi:10.1175/JCLI-D-13-00177.1, 2014.*

*Intergovernmental Panel on Climate Change (IPCC), Stocker, T.F., D. Qin, G.-K. Plattner, M. Tignor, S.K. Allen, J. Boschung, A. Nauels, Y. Xia, V. Bex and P.M. Midgley (eds.)]: Climate Change 2013: The Physical Science Basis. Contribution of Working Group I to the Fifth Assessment Report of the Intergovernmental Panel on Climate Change, Cambridge University Press, Cambridge, United Kingdom and New York, NY, USA, 1535 pp., doi:10.1017/CBO9781107415324, 2013.*

*Jones, C., Robertson, E., Arora, V., Friedlingstein, P., Shevliakova, E., Bopp, L., Brovkin, V., Hajima, T., Kato, E., Kawamiya, M., Liddicoat, S., Lindsay, K., Reick, C. H., Roelandt, C., Segschneider, J. and Tjiputra, J.: Twenty-first-century compatible co2 emissions and airborne fraction simulated by cmip5 earth system models under four representative concentration pathways, J. Clim., 26(13), 4398–4413, doi:10.1175/JCLI-D-12-00554.1, 2013.*

*Matthews, H. D.: Implications of $CO_2$ fertilization for future climate change in a coupled climate-carbon model, Glob. Chang. Biol., 13(5), 1068–1078, doi:10.1111/j.1365-2486.2007.01343.1, 2007.*

*Orr, J. C., Aumont, O., Yool, A., Plattner, K., Joos, F., Maier-Reimer, E., Weirig, M. -F., Schlitzer, R., Caldeira, K., Wicket, M., and Matear, R.: Ocean $CO_2$ Sequestration Efficiency from 3-D Ocean Model Comparison, in Greenhouse Gas Control Technologies), edited by Williams, D., Durie, B., McMullan, P., Paulson, C., and Smith, A., CSIRO, Colligwood, Australia, pp. 469-474, 2001.*

*Orr, J. C.: Modelling of ocean storage of $CO_2$ - The GOSAC study, Report PH4/37, IEA Greenhouse gas R&D Programme, 96pp., 2004.*

*Peters, G.P., Andrew, R. M., Boden, T., Canadell, J. G., Ciais, P., Le Quéré, C., Marland, G., Raupach, M. R., and Wilson, C.: The challenge to keep global warming below 2[deg]C, Nat. Clim. Chang., 3(1),4-6, doi:10.1038/nclimate1783, 2013*

*Zickfeld, K., Eby, M., Weaver, A. J., Alexander, K., Crespin, E., Edwards, N. R., Eliseev, A. V., Feulner, G., Fichefet, T., Forest, C. E., Friedlingstein, P., Goosse, H., Holden, P. B., Joos, F., Kawamiya, M., Kicklighter, D., Kienert, H., Matsumoto, K., Mokhov, I. I., Monier, E., Olsen, S. M., Pedersen,*

*J. O. P., Perrette, M., Philippon-Berthier, G., Ridgwell, A., Schlosser, A., Schneider Von Deimling, T., Shaffer, G., Sokolov, A., Spahni, R., Steinacher, M., Tachiiri, K., Tokos, K. S., Yoshimori, M., Zeng, N. and Zhao, F.: Long-Term Climate Change Commitment and Reversibility: An EMIC Intercomparison, J. Clim., 26(16), 5782–5809, doi:10.1175/JCLI-D-12-00584.1, 2013.*

---

## Author Response (AR1)

***Responses in Italic***

*First of all, the authors thank the reviewer very much for his thoughtful and constructive comments and advice.*

The authors investigate the impacts of ocean carbon injection (and of direct carbon capture and storage with no leakage) on the carbon inventories of the atmosphere, the ocean, and the land biosphere using the UVIC model. This is a solid study that should be published after taking into account the following comments:

1) The authors evaluate the impact of climate change on the fraction retained by comparing their complete mitigation (CM) simulations without emission forcing after 2020 and the RCP8.5 simulations with continued emissions (WE) (Line 181). They conclude (line 182) that larger climate change in RCP8.5 leads to a higher fraction of injected carbon retained in the ocean (FR).

I doubt that the difference between the CM and RCP85-WE simulations is indicative of climate change. I suspect that the higher fraction retained in the CM compared to the WE simulation is largely the result of differences in the Revelle factor/carbonate chemistry. The higher carbon emissions under RCP8.5 lead to a higher atmospheric and oceanic $CO_2$ and a higher Revelle factor. In turn a smaller fraction of anthropogenic carbon ends up in the ocean in the RCP8.5 case compared to the zero emission CM case. As in the long run, both simulations with and without ocean injection tend to achieve the same carbon partitioning between the ocean and the atmosphere (when neglecting ocean-sediment and weathering fluxes as done here) this mechanisms also affects the fraction retained. More injected carbon remains in the ocean for the low than for the high emission case.

A proper evaluation of the climatic impacts would require RCP8.5 simulations with carbon emissions, but with radiative forcing from anthropogenic agents set to zero. Then, climate would remain at equilibrium while atm. CO2 and carbonate chemistry would still change.

(Alternatively, I may misunderstand the experimental protocol. This would then require a clarification in the method section.).

*We thank the reviewer for this very important comment. The description of the diagnostic marker tracer in the experimental design section was insufficient, which led to the misunderstanding.*

*In lines 101 to 110 of the submitted manuscript, we describe how we deal with the injected carbon in the model. First, injected carbon is added to the total DIC pool of the model. Second, and in order to track the physical transport of injected $CO_2$ and its transport pathways from the individual injection sites, injected carbon is added to seven site-specific diagnostic 'marker tracer'. At the sea surface, these tracers have an instantaneous gas exchange with the atmosphere, i.e. as soon as some of the injected carbon reaches an ocean surface grid box, the value of the marker tracer in this surface ocean grid box is set to zero. The fraction retained computed from this tracer approach thus provides a lower limit estimate of carbon stored to carbon injected.*

*Hence, the Revelle Factor does not come into play with respect to the fraction retained. Differences in the fraction retained between the WE and CM simulations [section 3.3] cannot be explained by changes in the Revelle-Factor related to the invasion of anthropogenic $CO_2$ into the ocean, but only by climate induced changes of ocean circulation and stratification.*

*We apologize for the insufficient description of the diagnostic marker tracer in the original manuscript but have improved this now in the experimental design section (lines 123:130). The new text reads:*

*"To track the physical transport of the injected $CO_2$ and its transport pathways from the individual injection sites, injected carbon is added to seven site-specific diagnostic marker tracers. At the sea surface, we assume that these tracers have an instantaneous gas exchange with the atmosphere, i.e., as soon as the injected carbon reaches an ocean surface grid box, the value of the marker tracer in this surface ocean grid box is set to zero. The residence time of the injected $CO_2$ computed from this tracer approach (i.e. fraction retained, see below) thus, provides a conservative estimate of carbon stored to carbon injected, as it is unlikely that all of the injected carbon would instantly leave the ocean upon reaching a depth of 50 m. Furthermore, the fraction retained is not affected by changes in the Revelle Factor related to the invasion of anthropogenic $CO_2$ into the ocean."*

*In contrast to the fraction retained that counts only the injected carbon atoms (lines 125-129), the net fraction stored accounts for all potential feedbacks of carbon fluxes into and out of the ocean in response to the injection of $CO_2$ into the ocean (lines 130-135) and thus considers changes in the Revelle Factor in the surface ocean grid box. Our comparison of the net fraction stored with a lower estimate fraction retained is hence somewhat biased. We have performed a first test run for I-800 with a realistic gas exchange of the injected carbon at the ocean surface. The gas exchange of each individual marker tracer is computed by scaling the difference of the gas exchange of model DIC (including injected carbon reaching the sea surface) and a hypothetical gas exchange value considering a DIC value diminished by the sum of marker tracers to the individual marker tracer concentration. Thus, this approach does consider effects on the fraction retained through changes in the Revelle Factor. By comparing the fraction retained of I-800 as given in section 3.3 (Table 1) with the one of the realistic gas exchange simulation, we find that the latter increases by about 5% at the end of the injection period (year 2120). Consequently, the difference of the fraction retained and the net fraction stored in I-800 (Fig. 4 a) would increase, when assuming a realistic gas exchange of the injected carbon in the ocean surface grid boxes.*

2) A caveat of this study is that ocean sediments and the effect of calcium carbonate dissolution (also known as calcium carbonate compensation) are not considered. This caveat should be addressed in the introduction and conclusion section. This mechanisms could be relatively important as ocean carbon injection may bring the excess carbon close to deposits of calcium carbonate and thus would permit carbonate dissolution to occur on much faster time scale than for emissions into the atmosphere.

*Yes, we agree with the reviewer that this could be of importance. We have therefore clarified that we do not investigate the effect of calcium carbonate sediments feedbacks in our direct $CO_2$ injection experiments by running the model with and without a sediment sub-model. However, we feel that this issue should be discussed in the experimental design and conclusion sections. The new text in the experimental design section (lines 120:123) reads:*

*"Furthermore, we do not investigate the effect of CaCO₃ sediments feedbacks in our experiments, although the dissolution of CaCO₃ sediments near or downstream of an injection site is expected to reduce outgassing and increase the residence time of the injected CO₂ [Archer et al., 1998]."*

*The new text in the conclusion section (lines 468:470) reads:*

*"The neglect of the effect of the dissolution of CaCO₃ sediments near or downstream of the injection sites (see section 2.2) may have led to an underestimation of the FR and netFS in our injection experiments. The impact of this process would presumably be largest in the Atlantic due to the lower abundance of CaCO₃ sediments in the Pacific and Indian Ocean."*

3) The marker tracer used to compute the fraction retained should be explained in detail in the method section. As the fraction retained (FR) is a central metric in this study, it is not enough to refer to the literature.

*We agree with the reviewer and, as mentioned above, we have added a complete and detailed description of the marker tracer in the experimental design section.*

**With respect to further comments**

line 44: "reach a chemical equilibrium (mainly an equilibrium between the ocean and atmospheric carbon reservoirs)." This statement is not completely true as carbonate compensation and weathering feedbacks are important for time scales longer than 5000 years.

*Thank you for your careful reading. Carbonate compensation and weathering feedbacks have to be mentioned in this context as well and have been added to the revised manuscript. The new text in the introduction section (lines 46:48) reads:*

*"... reach a chemical equilibrium (mainly an equilibrium between the oceanic and atmospheric carbon reservoirs, although carbonate compensation and weathering feedbacks start acting on time scales longer than 5,000 years [e.g., Zeebe, 2012])."*

L 93: What about non-CO₂ forcings?

*This is a very good point. We have mentioned this in the experimental design section (lines 107:108). The new text reads:*

*"Note that non-CO$_2$ greenhouse gases and anthropogenic aerosol forcing agents as well as emissions from land-use change are not considered in our simulations."*

Line 127: could you please say a few more words about the diagnostic marker tracer. How is carbonate chemistry and air-sea and air-land flux computed for this tracer?

*A detailed description of the marker tracer has been added in the revised manuscript.*

Line 183: I doubt that the FR remains higher with than without climate change. I also doubt that this statement applies to all time scales (longer than the simulations).

*As mentioned above, in our simulations the Revelle Factor is neglected with respect to the fraction retained. Hence, differences in the fraction retained between the WE and CM simulations can, in our case, only be explained by a decrease of the ocean circulation and an increase of the ocean stratification as climate change progresses [Jain and Cao, 2005]. Consequently, and in line with our results (Table1) the fraction retained has to remain higher in the WE simulations compared to the CM runs. Figure R1 below illustrates that the fraction retained stays constantly higher in the I-800 WE simulation compared to the I-800 CM run over an extended time period of 1000 years (year 4020). Furthermore, we conducted an additional simulation forced under the RCP 8.5 emission scenario, but this time the CO$_2$-related radiative forcing is kept constant at pre-industrial level (i.e., I-800 no radiative forcing, Fig. R1). Its fraction retained stays below the ones of the I-800 WE and CM simulations. Unfortunately, the I-800 no radiative forcing simulation can only be compared until the year 3769. The results show clearly, that the I-800 CM and I-800 no radiative forcing runs converge with time as the hysteresis effect of climate change in the I-800 CM run keeps diminishing (Fig. R1).*

[Figure]

**Figure R1**: *Fraction retained for I-800 of the WE simulations (I-800 WE, black line) and for I-800 of the CM simulations (I-800 CM, blue line) until the year 4020. The red line illustrates the fraction retained for I-800 with the CO$_2$-related radiative forcing being kept constant at pre-industrial level (I-800 no radiative forcing) until the year 3769.*

**References:**

*Archer, D., Kheshgi, H., & Maier-Reimer, E.: Dynamics of fossil fuel CO$_2$ neutralization by marine Ca-CO3. Global Biogeochemical Cycles, 12(2), 259–276, doi:10.1029/98GB00744, 1998.*

*Jain, A. K. and Cao, L.: Assessing the effectiveness of direct injection for ocean carbon sequestration under the influence of climate change, Geophys. Res. Lett., 32(9), L09609, 2005.*

*Zeebe, R. E.: History of Seawater Carbonate Chemistry, Atmospheric CO 2 , and Ocean Acidification, Annu. Rev. Earth Planet. Sci., 40(1), 141–165, doi:10.1146/annurev-earth-042711-105521, 2012.*

***Responses in Italic***

*First of all, the authors thank Prof. Christoph Heinze very much for his thoughtful and constructive comments and advice. Note that new figures are shown at the end of this document.*

The manuscript investigates the effect of direct oceanic water column $CO_2$ injection on the redistribution of carbon under a high emission scenario following RCP8.5 its extension to 2300/2500 according to Meinshausen et al. (2011) and keeping emissions at a constant value until year 3020. The authors employ an Earth system model of intermediate complexity (UVic EMIC) and a standard protocol for prescribing the $CO_2$ injections. The study goes beyond the state-of-the-art by confronting not only an ocean biogeochemical model (with atmospheric reservoir) but a coupled Earth system model including also a terrestrial biosphere component (and a simple atmosphere representation) with ocean $CO_2$ injections. The model runs are carried out in a technically correct way as far as one can judge from the description. If I am not mistaken, the main result of the study is the following: $CO_2$ injection does not change the control run result for land carbon storage in a significant way for the forcing and injection protocol as applied. The last sentence in the conclusions (l. 348-350) maybe true in general but is hardly backed up by this particular study. The CMIP5 inter-model spread in land carbon storage change is much larger at year 2100 (Jones et al., J.Clim., 2013) than the amount discussed here as caused by ocean injection of $CO_2$. The manuscript confirms previous studies: A part of the injected $CO_2$ will outgas at a certain point in time, leading to less than 100% efficiency of the injection with respect to keeping anthropogenic excess $CO_2$ isolated from the atmosphere.

*Yes, we agree with the reviewer that the universality of the last concluding sentence is not completely backed up by our study. This would have required the comparison of the injection simulations with and without the land module. We have rephrased the last sentence of the conclusion (lines 489:491), accordingly. The new text reads:*

*"Nevertheless, our findings point to the importance of accounting for all carbon fluxes in the carbon cycle and not only for those of the manipulated reservoir, to obtain a comprehensive assessment of direct oceanic $CO_2$ injection in particular and carbon sequestration in general".*

*The comment related to the CMIP5 inter-model spread in land carbon storage change is discussed below.*

The authors correctly motivate their study with the current discussion on feasible mitigation targets to limit radiative warming to 2deg or 1.5deg C with respect to the pre-industrial. Respective emission scenarios would require at some point negative emissions. Why did the authors choose the business as usual strong warming scenario for their study? The amount of injected $CO_2$ is small in view if the $CO_2$ emissions in the RCP8.5 emission driven case. A more modest emission scenario would have been maybe more appropriate in view of the amount of injected $CO_2$ as used here.

*Our choice of the experimental design is motivated by the current trend of $CO_2$ emissions, which continues to follow largely the trajectory of the RCP 8.5 emission scenario [Peters et al., 2013] and also by our choice of the objective of our study, i.e., to investigate the response of the global carbon cycle during and after the direct $CO_2$ injections, considering a strong perturbation of the climate system. This has helped to investigate the effect of climate-induced changes on the fraction retained by comparing our 'Complete Mitigation runs' with the 'With Emissions simulations' (section 3.3). The justification of the small injection rate is that we wanted to compare and validate the fraction retained as well as the changes in seawater chemistry to the results of Orr et al. [2001; Orr, 2004].*

The terrestrial carbon cycle model used here is originally based on TRIFFID. This model has at times shown a more sensitive behavior to forcing than other models (see e.g. Friedlingstein et al., J. Clim., 2006/C4MIP, where both the Hadley Centre model and the UVIC model show significant outgassing after 2050). Would results with other terrestrial modules potentially show an even smaller deviation from the control run for the injection scenarios? The spread among different terrestrial carbon cycle modules concerning $CO_2$ uptake in Earth system models is large, also in view of the effect of nitrogen

cycle perturbations. The fluxes as presented in the paper should have been discussed in view of also these uncertainties. The authors correctly mention the as yet difficult to quantify $CO_2$ fertilization effect on land as large source of uncertainty.

*Yes, the authors agree that it is necessary to address and discuss the uncertainties related to the response of the terrestrial carbon cycle model to the direct $CO_2$ injections.*

*The process of $CO_2$ fertilization, which is here one of the dominant terrestrial carbon cycle feedbacks after $CO_2$ is injected, has direct relevance for the future trajectory of atmospheric $CO_2$ [IPCC, 2013] and thus for our targeted atmospheric carbon reduction of 70 GtC by the year 2120. The future strength of $CO_2$ fertilization in response to continued carbon emissions as in the 'With Emissions runs' is subject to the choice of the $CO_2$-fertilization parameterization and hence uncertain. In the new manuscript version we analyze the sensitivity of the $CO_2$-fertilization parameterization to the targeted atmospheric carbon reduction through direct $CO_2$ injections based on additional model runs, following the approach of Matthews [2007]. For these runs, we scaled the $CO_2$ sensitivity of the terrestrial photosynthesis model and have performed simulations of the RCP 8.5 control run, I-800 and I-3000, in which we have varied the strength of the $CO_2$ fertilization effect by increasing and decreasing it by ± 50% ($CO_2$ fert. high / low) relative to the default model. We have added a description of these simulations to the experimental design section (line 165:191). The new text reads:*

*"As mentioned in the introduction, this modelling study of direct $CO_2$ injection into the deep ocean is the first one to include a land component in order to assess, in addition to the atmospheric and oceanic carbon reservoirs, the long-term response of the terrestrial carbon pool to the targeted atmospheric carbon reduction through direct $CO_2$ injections. Since there is a significant amount of uncertainty in how the terrestrial system responds to changing atmospheric $CO_2$ concentrations [Friedlingstein et al., 2006], we have chosen to conduct several simulations with different terrestrial parameter values, i.e., a perturbed parameter study, to better understand how the terrestrial system could potentially respond to and affect the carbon cycle during deep ocean $CO_2$ injections. The*

*parameterization that we investigate is the $CO_2$ fertilization effect. The process of $CO_2$ fertilization is thought to stimulate terrestrial carbon uptake [e.g., Matthews, 2007]. This negative carbon cycle feedback results in reduced atmospheric $CO_2$ concentrations, and has likely accounted for a substantial portion of the historical terrestrial carbon sink [Friedlingstein et al., 2006]. Accordingly, it has direct relevance for the future trajectory of atmospheric $CO_2$ [IPCC, 2013] and thus for our targeted atmospheric carbon reduction of 70 GtC by the year 2120. However, the future strength of $CO_2$ fertilization in response to changing $CO_2$ is highly uncertain [e.g., Friedlingstein et al., 2006; Arora et al., 2013; Jones et al., 2013; Schimel et al., 2015]. In order to better quantify the role of $CO_2$ fertilization in the targeted atmospheric carbon reduction in the With Emissions simulations (section 3.4.3), we vary the $CO_2$ fertilization parameterization following the approach of Matthews [2007]. Thereby, we scale the $CO_2$ sensitivity of the terrestrial photosynthesis model by ± 50% ($CO_2$ fertilization = high / low) for repeated simulations that are otherwise identical to the RCP 8.5 control, I-800 and I-3000 runs. These variations scale the default strength of an increase in atmospheric $CO_2$ increase relative to pre-industrial levels that is used to calculate all processes in the canopy and leaf routines within the terrestrial photosynthesis model, leading to a respective increase or decrease in terrestrial gross primary productivity. This is achieved by adding the multiplicative parameter 'CO2_fert_scale' in the routine of the photosynthesis model and setting it to 1.5 for an increase of the $CO_2$ fertilization effect and to 0.5 for a respective decrease.*

*Hereafter, the perturbed control runs are referred to as RCP 8.5 control$_{CO2\_fert\_high}$ and RCP 8.5 control$_{CO2\_fert\_low}$. The perturbed injections runs are denoted as I-800 $_{CO2\_fert\_high}$, I-800$_{CO2\_fert\_low}$, I-3000 $_{CO2\_fert\_high}$ and I-3000$_{CO2\_fert\_low}$. We did not perform an I-1500 run because an ocean deep convection event that occurred after the injection period (see section 3.4.2) would make it too difficult to evaluate the results. No additional spin-up is needed; since the $CO_2$ fertilization effect only happens when atmospheric $CO_2$ concentration begins to increase, e.g., from the pre-industrial period onward."*

*In the results and discussion section (section 3.1), we describe carbon budgets of the perturbed control runs (RCP 8.5 control$_{CO2\_fert\_high}$ and RCP 8.5 control$_{CO2\_fert\_low}$) and how these differ from the unperturbed control run. In addition, we illustrate the results in a new Figure 2, in which, in addition to time series of all control runs, we also show bar diagrams of the absolute changes in the carbon reservoirs and fluxes between the perturbed control simulations and the unperturbed control run for the years 2120 and 3020. The new text in the results and discussion section (3.4.1, lines 218:236 ) reads:*

*"As expected, simulated terrestrial carbon uptake is higher in the RCP 8.5 control$_{CO2\_fert\_high}$ simulation because NPP is higher (not shown), when compared to the standard RCP 8.5 control run, resulting in a percentage increase in terrestrial carbon of about 5% in the year 2120 and of about 3% at the end of the simulation (Figs. 2 i, j). However, terrestrial carbon uptake declines more rapidly than in the control run, which is due to a faster saturation of the $CO_2$ fertilization effect as well as higher soil respiration. Consequently, the terrestrial biosphere switches about 20 years earlier to a stronger net carbon source (year 2121) before leveling off at very little net exchange between the terrestrial reservoir and the atmosphere after about year 2280 as occurring in the standard control run (Fig. 2 i).*

*Accordingly, the atmospheric carbon concentration in the RCP 8.5 control$_{CO2\_fert\_high}$ is lower, when compared to the RCP 8.5 control run, although the trends are similar (Figs. 2 a, b). Compared to the extended RCP 8.5 control run, the extended RCP 8.5 control$_{CO2\_fert\_high}$ ends with about 1% less atmospheric carbon (Figs. 2 a, b). The lower atmospheric carbon content in the RCP 8.5 control$_{CO2\_fert\_high}$, caused by the higher $CO_2$ fertilization effect, leads initially to a reduced carbon flux from the atmosphere to ocean (Fig. 2 c). By the year 2075, the carbon flux from the atmosphere to ocean is slightly higher, when compared to the control run, as the carbon flux from atmosphere to land starts to decrease with increasing $CO_2$ emissions (Fig. 2 d, g). Thus, total oceanic carbon in the control$_{CO2\_fert\_high}$ run stays below that of the control run with a percentage decrease of about 0.07% at the year 2120 and about 0.05% at the end of the simulation (Figs. 2 e, f).*

*Global carbon cycling in the RCP 8.5 control$_{CO2\_fert\_low}$ shows a similar response, although of opposite sign and higher magnitude (Fig. 2), which is for instance reflected in a percentage decrease in total land carbon of about 10% in the year 2120 and about 7% at the end of the simulation, when compared to the control run (Figs. 2 i, j). This is caused by the decreased $CO_2$ fertilization effect, which results in less NPP and thus in lower soil respiration."*

*Further, in a new section (3.4.3), we show how the carbon budgets of the perturbed injections runs (I-800 $_{CO2\_fert\_high}$, I-800$_{CO2\_fert\_low}$, I-3000 $_{CO2\_fert\_high}$ and I-3000$_{CO2\_fert\_low}$), when compared to the respective control runs, differ from the anomalies of the injection runs of our original 'With Emissions simulations'. We further present the difference for each carbon reservoir and flux at the year 2120 and 3020 in a new figure (new Fig. 6). For that purpose, we define error bars, which are for instance defined as the difference of the absolute changes in atmospheric carbon I-800 $_{CO2\_fert\_high}$ and $_{low}$ and the respective control runs and the absolute change in atmospheric carbon between I-800 and the control run of the 'With Emissions simulations'.*

*Finally, we discuss the terrestrial response to injections in the un- and perturbed runs in the context of the large uncertainty range related to the inter-model spread in future land carbon storage change [e.g., Arora et al., 2013; IPCC, 2013; Hajima et al., 2014]. We particularly discuss this in relation to the issue of nutrient limitation of photosynthesis currently missing in many terrestrial carbon cycle modules. There is high confidence that low nitrogen availability will limit land carbon uptake. Models that combine nitrogen limitation with rising $CO_2$ as well as changes in temperature and precipitation, predict a larger increase in projected future atmospheric $CO_2$ for a given $CO_2$ emission scenario [IPCC, 2013]. Models including terrestrial nutrient limitation are likely subject to a smaller terrestrial response to direct $CO_2$ injections into the deep ocean.*

*The new results and discussion section (3.4.3, lines 424:461) reads:*

*"Here we show how varying the $CO_2$ fertilization parameterization in the perturbed injection runs (i.e. i.e. I-800$_{CO2\_fert\_high\ and\ low}$ and I-3000$_{CO2\_fert\_high\ and\ low}$) changes carbon cycling and the leakage*

*of injected $CO_2$, when compared to the standard I-800 and I-3000 experiments of the With Emissions simulations.*

*As illustrated by the error bars in Figure 6 c, varying the $CO_2$ fertilization effect impacts the targeted atmospheric carbon reduction in I-800 of the With Emissions experiments, leading to a difference of -0.5 GtC to 0.02 GtC in the year 2120 and of 0.4 GtC to 1.1 GtC in the year 3020. Absolute changes in total oceanic carbon are also rather insensitive in these simulations with differences of only about -0.7 GtC to 0.4 GtC (0.01 GtC to 0.3 GtC) in the year 2120 (3020) (Figs. 6 d, e). Accordingly, the difference in the net fraction stored (netFS) in I-800 lies between -1% and 0.5% (Fig. 6 b) at the respective times. The slight differences in the fraction retained in I-800 (between -0.2 % and 0.3% at the respective times) are due to a slightly different climate in the perturbed simulations, when compared to the standard With Emissions runs, which is caused by the different atmospheric carbon concentrations (Fig. 6 c).*

*Absolute changes in terrestrial land carbon uptake and total land carbon show the largest sensitivities to the scaled $CO_2$ fertilization effect in I-800 (Figs. 6 f, g). By the end of the injection period, the difference in total land carbon between I-800 and the RCP 8.5 control run, shows that this terrestrial response could result in almost the same or less carbon storage, depending on the scaling of the $CO_2$ fertilization parameterization (Fig. 6 g). Higher $CO_2$ fertilization, i.e. I-800$_{CO2\_fert\_high}$, leads to a higher carbon flux from the atmosphere to land than in I-800, which counteracts the lower $CO_2$ fertilization effect that occurs in the standard I-800 because of less atmospheric carbon, when compared to the RCP 8.5 control run [see section 3.4.1]. This results in more land carbon of about 1.1 GtC (Fig. 6 g). The opposite is true for I-800$_{CO2\_fert\_low}$, leading to less land carbon by about 0.4 GtC in the year 2120, when compared to the difference between I-800 and the RCP 8.5 control run. By the end of the simulation, the perturbed injection simulation I-800$_{CO2\_fert\_high}$ has about 0.4 GtC less land carbon, relative to the difference of I-800 and the control run, which is caused by a slightly stronger cooling effect, because there is less atmospheric carbon than in I-800 (Fig. 6 g). This cooling also results in less*

*soil respiration. I-800$_{CO2\_fert\_low}$ has about 1.3 GtC less land carbon at the end of the simulations, when compared to the absolute change between I-800 and the respective control run. This can be explained by the reduced $CO_2$ fertilization effect that has led to a decreased NPP and consequently to a reduced soil respiration, when compared to I-800.*

*The magnitude of the responses that can be seen in the perturbed injection runs I-3000$_{CO2\_fert\_high}$ and I-3000$_{CO2\_fert\_low}$ are similar as in the perturbed I-800 runs.*

*Although the above response is informative, the future strength of the $CO_2$ fertilization effect also depends on other factors, such as water and nutrient availability [IPCC, 2013], which may be poorly simulated by our model. A key update since the Fourth Assessment Report by the IPCC is the implementation of nutrient dynamics in some of the CMIP5 land carbon models, such as in the NORESM-ME and CESM1-BGC models [Arora et al., 2013; Hajima et al., 2014]. There is high confidence that low nitrogen availability will limit land carbon uptake. Models that combine nitrogen limitation with rising $CO_2$ as well as changes in temperature and precipitation, predict a larger increase in projected future atmospheric $CO_2$ for a given $CO_2$ emission scenario [e.g., IPCC, 2013, Hajima et al, 2014]. Models including terrestrial nutrient limitation would likely be subject to a smaller terrestrial response if direct $CO_2$ injections into the deep ocean occurred. Thus, the introduction of nitrogen limitation in the land component of the UVic model would presumably result in less total simulated land carbon, because of lower NPP and soil respiration throughout the simulation, when compared to the terrestrial response in the shallow injection run (I-800) or for delayed emissions."*

*Further, we added paragraphs that address these new results in the abstract, introduction and conclusion sections.*

The authors say that direct injection of $CO_2$ is presently in conflict with . . . international protocols/conventions. This is correct but may also be an understatement. Direct $CO_2$ injection has been abandoned as a mitigation option because its environmental risks are potentially large (see WBGU report, 2006, for a summary of related risks, http://www.wbgu.de/en/special-reports/sr-2006-the-future-oceans/). The injection protocol of OCMIP/GOSAC as applied in the study does not account for the potential of fast rising bubbles after $CO_2$ injection (e.g., Bigalke et al., Environ. Sci. Technol.,

2008). Deeper ocean environments are sensitive to small pH variations (e.g., Gehlen et al., Biogeosciences, 2014). These aspects should be discussed in order to avoid misunderstandings by non-expert readers.

*This is a very good point. We did not intend to trivialize the potential ecological risks of direct $CO_2$ injection into the deep ocean. We have added a paragraph in the revised introduction section that addresses this issue (lines 61:65). The new text reads:*

*"Modelling studies are also safer than actual experiments because the rapid changes in seawater chemistry that could occur if direct $CO_2$ injections were tested might potentially harm marine ecosystems. These risks may be especially high for deep-sea benthic environments such as cold-water corals and sponge communities, which are adapted to special living conditions and thus may have a low capacity to acclimatize to rapid pH changes in their environment [e.g. IPCC, 2005, WBGU, 2006; Gehlen et al., 2014]."*

*Further, we have added the neglection of fast rising $CO_2$ bubbles [IPCC, 2005; Bigalke et al. 2008] in the experimental design section (lines 119:120). The new text reads:*

*"Consequently, the formation of $CO_2$ plumes or lakes as well as the potential risk of fast rising $CO_2$ bubbles are neglected [IPCC, 2005; Bigalke et al., 2008]."*

The authors discuss a transient Southern Ocean fluctuation of their model on one hand, and the lack of realistic internal variability in the EMIC employed on the other hand. The strength of EMICs is their low demand for computational resources. They would be suited to carry out ensemble simulations with large

numbers of members. This advantage could have been used to assess the robustness of the results. Maybe these would have become more significant or different for slightly perturbed initial conditions in an ensemble simulation?

*We have discussed possibilities to discriminate the impact of the natural variability (the deep convection) from the impact of $CO_2$ injections, for instance, during the injection phase before the onset of deep convection, or based on curve fitting of results from the other experiments, which show no deep convection events in the Southern Ocean. We came to the conclusion that no correct answer can be given without an ensemble simulation. Although the authors agree that it would be interesting and useful to perform an ensemble simulation with different initial conditions in order to assess the robustness of the ocean deep convection events, we feel that further analysis of it is beyond the scope of this study, which focuses on the response of the global carbon cycle during and after the $CO_2$ injections. In the manuscript we thus prefer to address this issue as done in line 420, but have added a short discussion on the advantage of an ensemble simulation with respect to the reviewers comment (line 413:415). The new text reads:*

*"Furthermore, ensembles would allow one to assess of the robustness of the occurrence of ocean deep convection events, which might become more significant or different for slightly perturbed initial conditions."*

Deep injection of $CO_2$ could potentially accelerate neutralizing fossil fuel $CO_2$ by dissolution of CaCO3 from the sea floor. Usually, on a 1000-years-time scale, the negative carbon cycle feedback through CaCO3 sediment dissolution is not important but rather on a several 10,000 year time scale (Archer, J.Geophys.Res., 2005). Water column injection potentially could change this though injection in the deep Pacific, where injection would be most effective, CaCO3 sediment is scarce. Nevertheless this aspect would warrant discussion. Is the (presumably small) CaCO3 effect larger than the land biosphere effect discussed here?

*Yes, this is a very important point that we have added to the results and discussion section (3.4.1, lines 340:347). The new text reads:*

*"The neglected effect of the CaCO$_3$ dissolution feedback in our injection experiments [see section 2.2] introduces another uncertainty with respect to the response of the global carbon cycle to direct CO$_2$ injections. Model simulations by Archer et al. [1998] have shown that CaCO$_3$ dissolution is sensitive to direct CO$_2$ injections throughout the Atlantic, but that it leads to only a slight impact on atmospheric pCO$_2$. However, a slightly modified trajectory of atmospheric CO$_2$ may, for instance, further impact the terrestrial carbon pool and fluxes, and could result in different terrestrial responses as in our With Emissions simulations. However, the comparison on how the marine CaCO$_3$ sediments feedback would affect global carbon cycling to the injections experiments without CaCO$_3$ sediments is the subject of future work and beyond the scope of this particular study."*

**With respect to small details:**

Abstract, l. 17: An . . . feature are effects (conflict singular/plural)

*Thank you for your careful reading. We have corrected this mistake.*

I find the introduction of the acronyms CM, WE, DAC and GIC not helpful. One can spell the terms out (maybe in italics).

*Yes, we agree that this could be confusing. We have spelled these acronyms out in italics.*

l. 136: misplaced comma

*Thank you, we have corrected this mistake.*

l. 183: comma after simulations required

*Thank you, we have corrected this mistake.*

Figure 1: The small rectangles with injection sites are difficult to identify.

*Yes, we have thickened the black rectangles in Figure 1 to make them easier to identify.*

Figure S2 should be placed in the main section. It shows the small effects. I do not want to stay anonymous.

*Yes, we have included Figure S2 in the main text as Figure 5*

***References:***

[revised manuscript text omitted]